# Decentralized Policy Gradients for Optimizing Generalizable Policies in Multi-Agent Reinforcement Learning[*]

## Abstract

Parameter Sharing (PS) is a widely used practice in Multi-Agent Reinforcement Learning (MARL), where a single neural network is shared among all agents. Despite its efficiency and effectiveness, PS can occasionally result in suboptimal performance. While prior research has primarily addressed this issue from the perspective of update conflicts among different agents, we investigate it from an optimization standpoint. Specifically, we point out the analogy between PS in MARL and Centralized SGD (CSGD) in distributed learning and hypothesize that PS may inherit similar convergence and generalization issues as CSGD, such as lower convergence levels of key metrics and larger generalization gaps. To address these issues, we propose Decentralized Policy Gradients (DecPG), which leverages the principles of Decentralized SGD. We use an environment with additional perturbations injected into the observation and action spaces to evaluate the generalization of DecPG. Experiments are conducted on homogeneous scenarios from MPE Simple Spread and SMAC, using the vanilla SGD optimizer. Empirical results show that DecPG outperforms its centralized counterpart, PS, across various aspects—achieving higher rewards, smaller generalization gaps, and flatter reward landscapes. The results confirm that PS suffers from convergence and generalization issues similar to those of CSGD, and show that our DSGD-based method, DecPG, effectively mitigates these problems—offering a new optimization perspective on MARL algorithm performance.

## 1 Introduction

In cooperative Multi-Agent Reinforcement Learning (MARL), multiple agents interact cooperatively with each other and with the environment to achieve a common goal. Each agent has an individual policy, and together they form a joint policy. MARL algorithms aim to train this joint policy to maximize the return. A typical practice to optimize the joint policy is Parameter Sharing (PS) (Foerster et al., 2016; 2018; Gupta et al., 2017; Rashid et al., 2020; Yu et al., 2022). Instead of training a separate neural network for each agent policy, PS employs a single shared network for all agents. It is simple to implement, and training one shared policy with data from all agents offers promising sample efficiency. Because of these advantages, PS has been adopted in various benchmark algorithms, such as MAPPO (Yu et al., 2022) and QMix (Rashid et al., 2020).

Interestingly, the PS in MARL shares great similarities with the centralized SGD (CSGD) used in distributed learning. In particular, distributed learning trains deep neural networks at scale by leveraging parallel training across multiple nodes. Each node maintains a copy of the initial model and has access to a local dataset, allowing it to perform gradient descent locally. In CSGD, there is a central parameter server that aggregates the updates from local nodes and broadcasts the aggregated results back to all nodes, corresponding exactly to the update style of PS. This approach is essentially equivalent to single-worker SGD, but its distributed nature allows CSGD to parallelize the computation across multiple nodes, enabling larger effective batch sizes and significantly improving training efficiency (Dekel et al., 2012).

---

[*]This paper used ChatGPT to assist with proofreading. All ideas, claims, and results are entirely developed by the authors.

This analogy between PS and CSGD intrigues us to investigate whether PS inherits similar optimization issues from CSGD. Particularly, in practice, CSGD often suffers from convergence to suboptimal loss levels or complete failure to converge with large batches (Jastrzebski et al., 2017; You et al., 2017; Zhang et al., 2019), and poorer generalization (Goyal et al., 2017; Keskar et al., 2017). PS has also been found to result in suboptimal performance in several experiments, including both heterogeneous and homogeneous agent settings (Kim & Sung, 2023; Qin et al., 2025). Most existing papers address the problem from only one perspective—by mitigating the conflicting updates among different agents, for instance, via improving the parameter sharing mechanism, as in SePS (Christianos et al., 2021) and SNP-PS (Kim & Sung, 2023), or utilizing sequential training, like HAPPO (Zhong et al., 2024) and A2PO (Wang et al., 2023). However, alternative perspectives, particularly from the connection between PS and CSGD, remain largely underexplored.

On the other hand, this connection can offer a novel understanding of the performance limitations of PS in MARL and provide insights to address them from an optimization standpoint. Crucially, in CSGD, one effective way to address the aforementioned problems is to shift from a centralized framework to a decentralized one, i.e., Decentralized SGD (DSGD) (Lian et al., 2017; Zhang et al., 2021; Zhu et al., 2023). Unlike in CSGD where parameter averaging is performed on a central server, in DSGD, parameters are averaged locally at each node within a neighborhood defined by a *communication topology*. Although the local models are initialized with identical parameters, partial averaging causes the local models to diverge. This divergence introduces an intrinsic noise that acts as a regularization, smoothing the loss landscape and stabilizing training (Zhang et al., 2021; Zhu et al., 2023). Theoretically and empirically, this noise has been shown to improve convergence and generalization relative to CSGD (Zhang et al., 2021; Zhu et al., 2023).

Inspired by these findings, this paper investigates whether incorporating DSGD in MARL could effectively address these issues caused by PS. To this end, we propose **Decentralized Policy Gradient (DecPG)**, which incorporates DSGD's style of update into a policy-based MARL algorithm. Since the original use case and theoretical analyses of DSGD are based on the assumption of homogeneity across nodes, we aim to maintain consistency with this setting by (1) focusing on homogeneous MARL tasks, (2) adopting Centralized Training Decentralized Execution (CTDE) (Amato, 2024; Foerster et al., 2018; Sunehag et al., 2018) to mitigate the non-stationarity inherent in MARL, and (3) using MAPPO (Yu et al., 2022) to constrain policy updates. We compare PS to DecPG with topologies from sparse to dense in the Multi-Agent Particles (MPE) Simple Spread (Mordatch & Abbeel, 2018) and StarCraft Multi-Agent Challenge (SMAC) (Samvelyan et al., 2019). We analyze how convergence and generalization behavior vary across these settings. Through comprehensive experiments, we empirically demonstrate that PS in MARL also suffers from suboptimal convergence and generalization. The DSGD-based method, DecPG, mitigates these problems and outperforms PS.

Our contributions are as follows.

- We propose DecPG—a policy gradient-based MARL algorithm built on DSGD—to improve convergence and generalization over its centralized counterpart, PS.

- We provide comprehensive empirical results to show that DecPG outperforms PS in training and test performance in most scenarios.

- We analyze the effect of communication topology sparsity—from sparse to dense—on final generalization performance. We evaluate a series of topologies whose spectral gaps span approximately linearly from 0 to 1, as defined by k-nearest-neighbor graphs. Our results show that the generalization capability tends to improve as the topology becomes sparser.

- We provide a qualitative visualization of the reward and loss landscapes for DecPG under topologies of varying sparsity, as well as for PS, showing that reward landscape smoothness improves from PS to DecPG as the topology becomes sparser.

## 2 Related Work

This paper investigates whether PS in MARL exhibits similar convergence and generalization problems as CSGD in distributed learning, and whether the proposed DSGD-based method, DecPG, can address these

issues. Accordingly, this section reviews the related work about the characteristics and issues of PS and CSGD, and the recent developments in DSGD.

**Parameter Sharing (PS)** PS employs a single neural network to represent the policies of all agents and is trained with trajectories collected from all agents, making it particularly efficient and scalable as the number of agents increases. With the inclusion of agent ID (usually a one-hot encoding) to the input, PS can handle not only homogeneous but also heterogeneous problems (Rashid et al., 2020; Yu et al., 2022). It has been widely used in benchmark algorithms and has demonstrated outstanding empirical performance (Foerster et al., 2018; Gupta et al., 2017; Rashid et al., 2020; Yu et al., 2022). Despite its empirical success, PS has been found to be suboptimal in certain scenarios, including both homogeneous and heterogeneous settings (Kim & Sung, 2023; Qin et al., 2025). One direction to address it is by modifying the parameter sharing strategy, typically by making the neural network partially shared and partially specialized, rather than fully shared. These methods include SePS (Christianos et al., 2021), SNP-PS (Kim & Sung, 2023), Kaleidoscope (Li et al., 2024), GradPS (Qin et al., 2025), etc. The other direction focuses directly on the MARL algorithm itself, such as HAPPO/HATRPO (Zhong et al., 2024) and A2PO (Wang et al., 2023), utilizing sequential agent updates instead of simultaneous ones. These existing approaches aim to mitigate conflicts between different agents' updates through algorithmic-level designs. In contrast, this paper focuses on improving generalization from an optimization perspective, a relatively unexplored direction in MARL.

**Centralized SGD (CSGD) and Decentralized SGD (DSGD).** In distributed learning, larger batch sizes are crucial for enhancing parallelism, but **CSGD** (or equivalently, single-worker SGD) is found to suffer from performance problems in large batch settings (Goyal et al., 2017; Jastrzebski et al., 2017; You et al., 2017; Zhang et al., 2019). Zhang et al. (2019) observe that using CSGD with batch sizes exceeding a certain threshold significantly deteriorates convergence, resulting in a high and non-decreasing training loss. This behavior is speculated to stem from a lack of stochasticity in the gradients. On the other hand, Goyal et al. (2017) discover an increasing trend in validation error with larger batch sizes, indicating a decline in generalization performance. Keskar et al. (2017) analyze this phenomenon from the perspective of optimization landscape flatness, where it is generally believed that flatter minima correspond to better generalization (Hochreiter & Schmidhuber, 1997). They empirically demonstrate that larger batch sizes tend to converge to sharper minima, whereas smaller batch sizes lead to flatter ones. **DSGD** was once considered a compromise to CSGD, viewed as useful only under poor bandwidth or high network latency. However, it has since been shown to outperform CSGD in large-batch training scenarios. Lian et al. (2017) prove that DSGD can achieve the same convergence rate as CSGD and exhibit a similar asymptotic linear speedup in computational complexity with respect to the number of nodes. This enables DSGD to converge in less wall-clock time than CSGD in practice, due to its significantly lower communication overhead on individual nodes. Koloskova et al. (2020) establish a unified framework that proves convergence guarantees for DSGD under different scenarios, including changing topologies, i.i.d. and non i.i.d. data distributions. Zhang et al. (2021) show that DSGD can achieve better convergence than CSGD due to the noise arising from differences among local model weights across nodes. Empirically, this noise is found to stabilize convergence and performs better than injecting random Gaussian noise into CSGD. Zhu et al. (2023) prove that DSGD is asymptotically equivalent to the average-direction sharpness-aware minimization algorithm Wen et al., 2022, revealing several important insights into its convergence and generalization behaviors. First, the decentralization-induced noise arising from partial averaging is landscape-dependent, implicitly penalizing sharp curvature in the loss surface and thereby encouraging flatter minima that yield better generalization. Second, this same noise introduces a gradient-smoothing effect, which stabilizes the optimization trajectory and can improve convergence. Third, unlike the stochastic-sampling noise in CSGD, the implicit regularization strength of DSGD is independent of batch size, suggesting that its generalization benefit persists even in large-batch settings. Ye et al. (2025) justify an upper bound for the generalization error in DSGD under data heterogeneity and without the assumption of bounded stochastic gradients. This upper bound reveals that a good generalization in DSGD is favored by lower stochastic noises and reduced data heterogeneity, which correspond to larger batch sizes and more i.i.d. data distributions across nodes. Deng et al. (2023) derive generalization bounds for DSGD that specifically depends on the spectral gap of the communication topology. They show that generalization errors increase as the spectral gap narrows, i.e., as the topology becomes sparser.

# 3 Decentralized Policy Gradient

**Preliminaries**   We model a cooperative multi-agent reinforcement learning (MARL) problem as a Decentralized Partially Observable Markov Decision Process (Dec-POMDP), defined as

$$\mathcal{M} = (\mathcal{S}, \{\mathcal{A}_i\}_{i=1}^n, P, R, \{\mathcal{O}_i\}_{i=1}^n, \{O_i\}_{i=1}^n, \gamma, n, \rho_0).$$

where $n$ denotes the number of agents, $\mathcal{S}$ is the (global) state space, and $\rho_0$ is the initial state distribution. The joint action space is $\boldsymbol{a} = (a^1, a^2, \dots, a^n) \in \prod_{i=1}^n \mathcal{A}_i$, where $\mathcal{A}_i$ is the action space of agent $i$. The environment transition dynamics are defined by $P(s' \mid s, \boldsymbol{a})$, which gives the probability of transitioning to state $s'$ given the current state $s$ and joint action $\boldsymbol{a}$. The reward function $R(s, \boldsymbol{a})$ provides a shared scalar reward $r$ to all agents. The discount factor is $\gamma \in [0, 1)$. Each agent $i$ receives a local observation $o^i \in \mathcal{O}_i$, where $\mathcal{O}_i$ is its observation space. The observation function $O_i(o^i \mid s)$ defines the probability of agent $i$ observing $o^i$ given the global state $s$. Let $\boldsymbol{o} = (o^1, \dots, o^n) \in \prod_{i=1}^n \mathcal{O}_i$ denote the joint observation. Assuming the joint policy $\boldsymbol{\pi}(\boldsymbol{a} \mid \boldsymbol{o}) = \prod_{i=1}^n \pi^i(a^i \mid o^i)$, the objective is to maximize the expected discounted cumulative reward $\mathbb{E}\left[\sum_{t=0}^\infty \gamma^t r_t\right]$.

**Decentralized Policy Gradient (DecPG)**   We propose a DSGD-based MARL algorithm, DecPG. To enable that, we simulate MARL as a distributed learning case. Every agent forms a local node with a copy of the initial model $\theta_0$, an independent dataset $D_i$, and a loss function $L_i$. At the beginning of every iteration, agents use their local models $\theta_i$ to roll out trajectories, forming the local dataset for training. The dataset for agent $i$ at $m$th iteration is defined as:

$$D_m^i = \{\tau_b^i\}_{b=1}^B,$$

where $B$ is the batch size, and each trajectory $\tau_b^i$ of length $T$ is defined as: $\tau_b^i = \{(o_t^i, s_t, a_t^i)\}_{t=0}^T$, where $s_0 \sim \rho_0(s)$, $o_t^i = O_i(s_t)$, $a_t^i \sim \pi^i(a_t^i \mid o_t^i)$, $s_{t+1} \sim P(s_{t+1} \mid s_t, a_t^i, \boldsymbol{a}_t^{-i})$, $\boldsymbol{a}_t^{-i} \sim \boldsymbol{\pi}^{-i}(\boldsymbol{a}_t^{-i} \mid \boldsymbol{o}_t^{-i})$.

The communication topology $P$ for DecPG is defined by a doubly stochastic $k$-Nearest Neighbor (kNN) graph. Figure 1 illustrates an example when $n = 10$. Intuitively, agents are arranged in a circle, and each agent is connected to its $k$ nearest neighbors. $P$ is an $n \times n$ matrix, where $P_{i,j}$ is the weight of the connection from agent $i$ to agent $j$. In a kNN topology, $P_{i,j} = 0$ if two agents are disconnected and $P_{i,j} = 1/(k+1)$ if agent $i$ is connected to agent $j$. For a scenario with $n$ agents, there are $n - 1$ possible kNN topologies. The case $k = 0$ corresponds to a completely disconnected graph, which is equivalent to non-parameter sharing. When $k = n - 1$, the graph is fully connected, corresponding to PS. In this paper, we consider values of $k$ ranging from 1 to $n - 2$, forming an approximately linear progression of spectral gap values from 0 to 1 (Figure 6 in Appendix A.2). The spectral gap is defined as $1 - |\lambda|$, where $|\lambda| = \max_{i \geq 2} |\lambda_i|$ is the second-largest modulus among eigenvalues $\lambda_i$ of $P$. It measures the rate at which information diffuses across the network. For instance, a smaller spectral gap implies a sparser topology. The spectral gap also influences the strength of the noise induced by DSGD (Zhu et al., 2023), with a smaller spectral gap corresponding to stronger noise.

During training, each agent adopts the same loss function but computes it using its own local dataset $D_m^i$, and the loss is computed as:

$$L_m^i = \frac{1}{B} \frac{1}{T} \sum_{b=1}^B \sum_{t=1}^T \ell(o_{b,t}^i, s_{b,t}, a_{b,t}^i),$$

where $\ell(\cdot)$ denotes either the policy or critic loss, and $(o_{b,t}^i, s_{b,t}, a_{b,t}^i)$ denotes $t$-th step in $b$-th trajectory from $D_m^i$.

After gradients being computed locally for all agents, the DSGD mixing is performed for every agent, over itself and its neighbours, defined as:

$$\theta_{m+1}^i = \sum_j^n P_{i,j} \theta_m^j - \eta \cdot \nabla L_m^i(\theta_m^i)$$

For instance, in a scenario with 10 agents and a topology with $k = 2$ (Figure 1), during the DSGD mixing step, the updated model weights of agent 0 are given by

$$\theta_{m+1}^0 = P_{0,9} \cdot \theta_m^9 + P_{0,0} \cdot \theta_m^0 + P_{0,1} \cdot \theta_m^1 - \eta \cdot \nabla L_m^0(\theta_m^0),$$

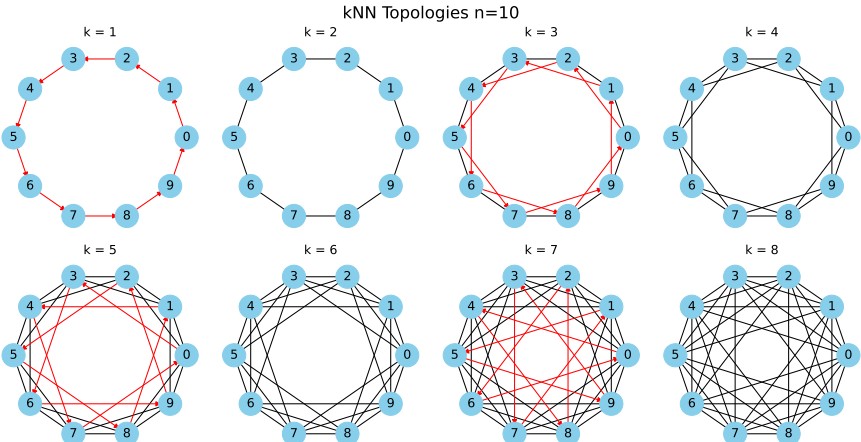

Figure 1: kNN communication topologies for $n = 10$ with $k = 1$ to 8. Black edges indicate bidirectional connections; red edges indicate unidirectional edges.

where $P_{0,9} = P_{0,0} = P_{0,1} = \frac{1}{3}$.

This makes DecPG contrary to PS, which averages the gradients of all agents, analogous to CSGD. The loss function of PS is as follows:

$$\theta_{m+1}^{PS} = \theta_m^{PS} - \eta \cdot \sum_{i=1}^{n} \nabla L_m^i(\theta_m^{PS})$$

In evaluation, the average model $\theta_m^{avg} = \frac{1}{n} \sum_i^n \theta_m^i$ is used to rollout, following standard practice in DSGD (Lian et al., 2017; Ye et al., 2025; Zhang et al., 2021). The theoretical advantages of DSGD established by Zhu et al. (2023) are also based on this averaged model.

To better understand the effects of DSGD in MARL, we implement DecPG to align as closely as possible with the DSGD framework (Zhu et al., 2023). Consequently, we address the key discrepancies between MARL and the DSGD setting as follows.

Unlike supervised learning, where the local datasets are stationary once defined, the local datasets for agents in MARL have a changing distribution. Each agent's experience distribution depends on its local policy as well as the influence of other agents' policies. To mitigate the effect of this distribution shift on DSGD, we use MAPPO (Yu et al., 2022) as the backbone because it enforces trust-region control via gradient clipping to make the change in agent policy reasonably small in a local temporal range. Furthermore, we employ the Centralized Training Decentralized Execution (CTDE) (Amato, 2024; Foerster et al., 2018; Sunehag et al., 2018) paradigm to mitigate the non-stationarity caused by inter-agent interaction. With these measures, we make the local data as reasonably stationary as possible.

The theoretical benefits of DSGD (Zhu et al., 2023) are established under the assumption of homogeneous data across nodes. Moreover, Ye et al. (2025) show that data heterogeneity can slow convergence and impair generalization, which could obscure the direct effects of DSGD. Therefore, we focus primarily on homogeneous settings, where agents are identical.

## 4 Experiments

We use Simple Spread from Multi-Agent Particle Environment (MPE) (Mordatch & Abbeel, 2018) and StarCraft Multi-Agent Challenge (SMAC) (Samvelyan et al., 2019) as the testbeds. Simple Spread is a straightforward task where agents navigate to reach landmarks while avoiding colliding with each other. The reward is easy to interpret, consisting of the agents' distances to landmarks and penalties for collision. We treat Simple Spread as a toy example for its simplicity and interpretability, and use it to identify

performance patterns across different algorithms. For Simple Spread, we use scenarios with agent counts ranging from 8 to 12.

On the other hand, SMAC is a widely adopted MARL benchmark featuring challenging micromanagement tasks, where agents control a team of units to defeat the enemies. The reward structure of SMAC is more complicated, involving factors such as ally and enemy health, unit losses, and the winning status. Due to this complexity, we primarily use SMAC to evaluate the overall performance of algorithms rather than to interpret fine-grained trends. In SMAC, we select the homogeneous maps *5m_vs_6m*, *10m_vs_11m*, *27m_vs_30m*, and *6h_vs_8z*, covering a variety of agent counts. These maps are considered challenging as they involve more enemies than agents.

DecPG is tested under different kNN communication topologies that approximately evenly span the spectral gap, ranging from 0 to 1. For instance, we select 4 topologies $k = 2, 4, 6, 8$ for $n = 10$, with spectral gaps around 0.13, 0.35, 0.63 and 0.89 respectively, roughly spanning the entire spectral gap linearly.

We compare DecPG with the standard PS baseline. Additionally, we include a PS variant with entropy regularization (PS+entropy) for comparison, as we observe that DecPG tends to induce more exploration reflected by higher policy entropy than standard PS (see Appendix A.3). The PS+entropy baseline is implemented by adding the average policy entropy to the loss function, scaled by a coefficient specified in Appendix A.1. The implementation follows MAPPO (Yu et al., 2022).

To facilitate the interpretation of results, we adopt vanilla SGD as the optimizer, ensuring consistency with its theoretical justification in (Zhu et al., 2023). Following the optimal hyperparameters suggested by Yu et al. (2022), all experiments are conducted using full batch training. We adopt a neural network architecture consisting of 3 linear layers, followed by a GRU layer, another linear layer, and finally an output layer, for both the actor and critic. For details on other hyperparameters, please refer to Appendix A.1.

To evaluate DecPG, we consider the following metrics. First, we examine the training performance (average episode reward), measured as the policy's performance on the same environment as in training, but evaluated with different random seeds and using the deterministic average policy. It is important to note that this evaluation is insufficient for assessing generalization, as the variation introduced by changing random seeds is often minimal. For example, in Simple Spread, different seeds do not change the distribution for the initialization of agent and landmark positions.

To test the generalization, we design a modified version of the training environment as the test environment, in which perturbations are added to agents' observations and actions. We inject Gaussian noise into the observation space, with a mean of 0 and a standard deviation equal to $c$ times the sample standard deviation. Since the action space is discrete for SMAC and Simple Spread, we assign a small probability $\varepsilon$ for agents' actions to be replaced with random ones uniformly sampled from each agent's set of available actions at each timestep. These modifications ensure that the test environment does not follow the exact same distribution as the training environment, while still preserving the core task, unlike some prior works that alter the number of agents, effectively turning the evaluation into a few-shot or zero-shot setting (Tian et al., 2023). A well-generalized policy is expected to exhibit minimal performance degradation under such perturbations. The test environment-related hyperparameters are shown in Appendix A.1.

We regard the test performance (test average episode reward) as the primary metric for the generalization power of a policy (Figure 3). Generalization gap is provided as an additional metric for generalization (Figure 4), defined as the difference between test and train performance. This gap reflects the extent to which performance degrades when transitioning from the training to the test environment. A higher test performance and a smaller generalization gap indicate stronger generalization.

Additionally, we visualize the reward and loss landscapes of the learned policies in 3D (Figure 8). This is done by selecting two orthonormal directions in the policy parameter space and perturbing the policy along those directions. These visualizations provide qualitative insight into generalization from the perspective of landscape flatness.

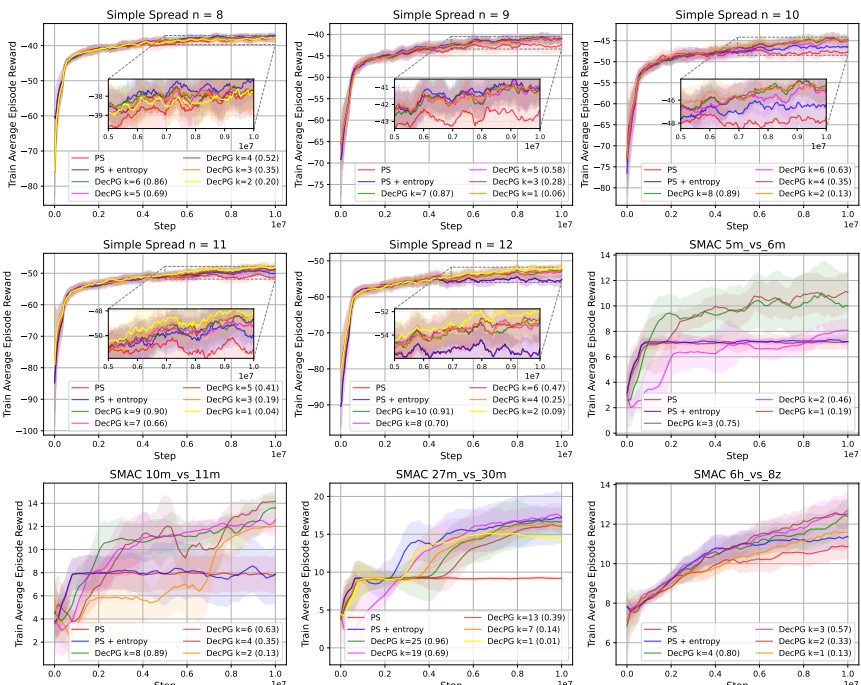

Figure 2: Training average episode rewards on MPE Simple Spread and SMAC, evaluated using deterministic policies. We compare PS, PS with entropy regularization (PS+entropy), and DecPG with different topologies (DecPG $k=x$). For DecPG, the value $x$ in parentheses indicates the spectral gap of the corresponding topology.

We also conduct supplementary analyses on metrics that quantify the decentralization-induced noise and on the performance of DecPG policies under different evaluation configurations. The corresponding results are presented in Appendix A.5 and Appendix A.6.

All experiments were run for 5 seeds, on Google TPU v4-8. The average episode rewards for training and testing are averaged over 32 and 1000 trials, respectively. Results shown in Figures 2, 3, and 4 are the median values. Shaded regions in Figure 2 display the 95% confidence zone. Bars in Figure 3 show the standard deviation.

## 5 Results and Findings

Through comprehensive empirical evaluation, we have the following findings.

### 5.1 DecPG achieves better convergence and exploration than the PS baselines during training.

Figure 2 shows the training average episode rewards obtained using the deterministic policy. Across all scenarios except Simple Spread n=8 and n=9, at least one topology of DecPG outperforms both PS and PS+entropy, and in the majority of scenarios (except for Simple Spread n=8, n=9, and *27m_vs_30m*), all DecPG algorithms surpass PS and PS+entropy, highlighting the convergence advantage of DecPG over the PS baselines.

By analyzing the evolution of policy entropy (Figure 7), we observe that DecPG's entropy decreases steadily, whereas PS exhibits a much faster drop. We deduce that the noise introduced by DecPG slows convergence, thereby encouraging more exploration. Entropy regularization helps PS achieve a similar level of exploration in most tasks; however, this effect does not lead to the same improvement in convergence as observed with DecPG.

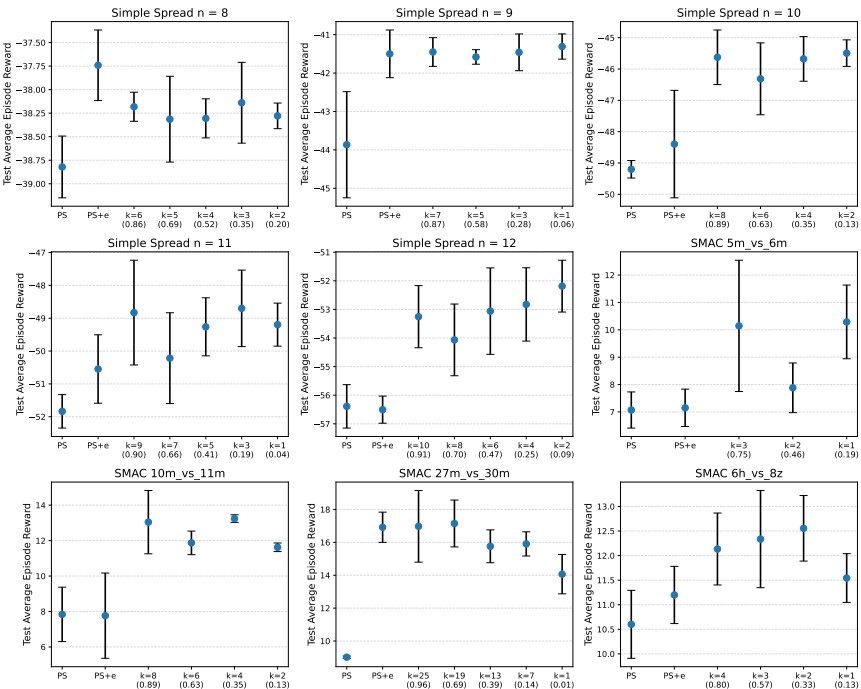

Figure 3: Test average episode rewards of the final deterministic policies on MPE Simple Spread and SMAC.

## 5.2 DecPG demonstrates better generalization than PS.

Figure 3 shows the test average episode rewards, evaluated using the final converged policies of the PS and DecPG baselines. In all scenarios, PS consistently performs the worst. Performing better than PS in most tasks, PS+entropy achieves the best test performance in the $n = 8$ scenario and performs comparably to DecPG in $n = 9$ and *27m_vs_30m*. However, in the rest of the tasks, it is outperformed by at least one DecPG policy.

Across all scenarios except Simple Spread $n = 8$, DecPG consistently outperforms the PS baselines in terms of test performance, highlighting the generalization advantage of DSGD-style updates over CSGD.

## 5.3 Relationship between DecPG performance and topology sparsity.

In terms of training rewards (Figure 2), we do not observe a consistent trend of performance difference among DecPG topologies across tasks. In most Simple Spread scenarios, all topologies perform similarly. While the sparsest topologies in Simple Spread n=11, n=12, and *5m_vs_6m* are the best performing DecPG policies, these are the worst performing topologies for DecPG in *6h_vs_8z*, *10m_vs_11m* and *27m_vs_30m*. Topologies with moderate sparsity generally outperform both PS and PS+entropy across most tasks. In particular, those with spectral gaps between 0.5 and 0.7 (corresponding to the second densest topology selected for each task, e.g. k=6 for *10m_vs_11m*) outperform PS and PS+entropy in all but Simple Spread n=8 and n=9.

Regarding the test performance (Figure 3), we observe three general trends: increasing (n=10, n=11, n=12, and *6h_vs_8z*), leveling (n=8, n=9, and *10m_vs_11m*), and decreasing (*27m_vs_30m*). It indicates that the behavior of different topologies of DecPG depends largely on the task itself. For instance, tasks like n=12 favor sparser topologies, implying that the underlying problem benefits more from stronger regularization on the optimization landscape's sharpness. In contrast, for *27m_vs_30m*, excessive regularization may hinder optimization, leading to a drop in test performance. Merging the findings from the three trends, a moderate level of topology sparsity—avoiding the two extremes—appears to be the safer and more robust choice in practice, aligning with the situation in training performance.

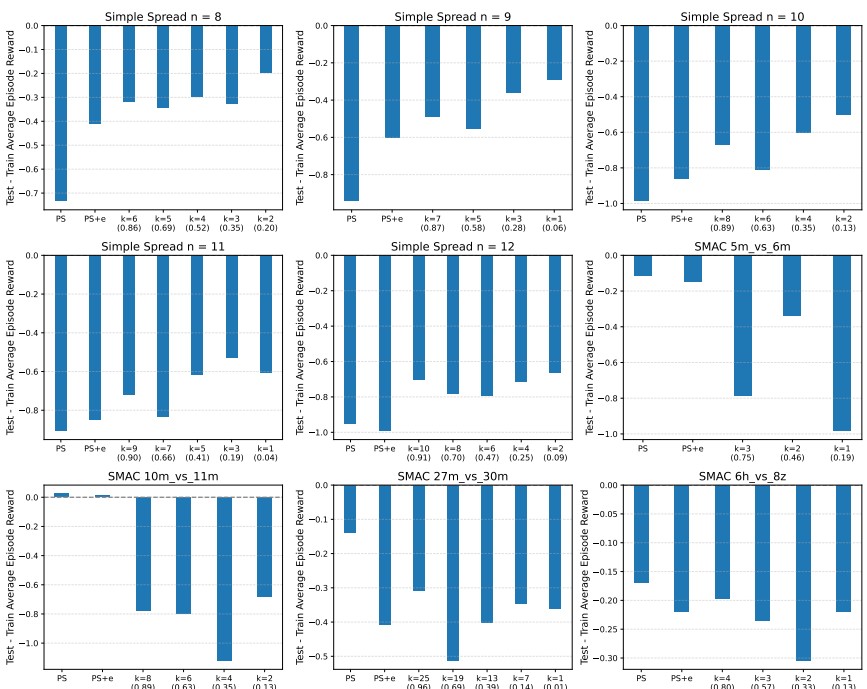

Figure 4: Generalization gap on MPE Simple Spread and SMAC, computed as the difference between test and training rewards.

Generalization gaps are shown in Figure 4. Across all Simple Spread scenarios, we observe a decreasing trend in generalization gaps as topology becomes sparser (i.e., as $k$ decreases), though minor fluctuations exist. This suggests that generalization improves with increased sparsity. This observation can be explained by the theoretical findings of Zhu et al. (2023), which show that sparser topologies in DSGD impose stronger sharpness regularization on the loss landscape. Therefore, they encourage convergence to flatter minima that are less sensitive to perturbations in the test environment.

However, we do not observe a similar trend in the SMAC scenarios (Figure 4). Instead, we find that the generalization gaps are more closely correlated with absolute test performance. Specifically, higher test performance is associated with larger generalization gaps. We attribute this to the nature of the selected SMAC maps, which involve imbalanced ally and enemy team sizes and demand more fine-grained coordination among agents to defeat larger numbers of opponents. As a result, perturbations in the test environment tend to have a more catastrophic impact on well-performing policies than on those that already perform relatively poorly. Although this effect dominates the generalization gap patterns and obscures the trend observed in Simple Spread, the test performance results presented earlier remain sufficient to reveal the generalization capabilities of DecPG under different topologies.

Overall, the comparison across different DecPG topology sparsities reveals that the optimal choice of sparsity varies significantly across tasks. In general, a small amount of regularization introduced by DSGD (represented by the densest topology) can already outperform the case without any regularization (represented by PS). However, to achieve robust training and test performance, a moderate level of topology sparsity tends to be more reliable. Empirically, we find that a spectral gap between 0.5 and 0.7 provides a robust choice across the evaluated tasks.

This observation may relate to the optimization–regularization trade-off discussed in Zhu et al. (2023), which suggests that sparser topologies introduce stronger noise and thus stronger regularization, but excessive regularization can dominate the optimization objective, leading to performance degradation.

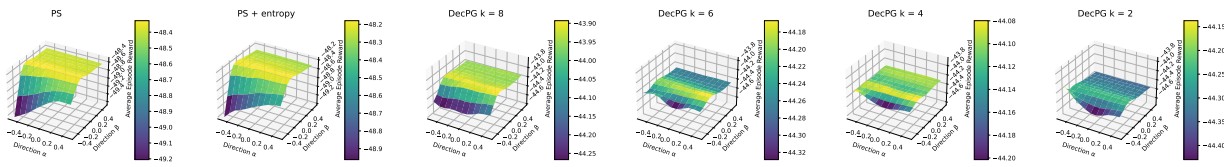

(a) Average episode reward landscape surface

Figure 5: Visualization of reward landscapes in MPE Simple Spread $n = 10$, for PS, PS with entropy regularization (PS+entropy), and DecPG with different topologies (DecPG k=x). The figures show 3D surfaces of the reward with respect to perturbations in two orthonormal directions of the parameter space.

### 5.4 DecPG achieves flatter reward landscapes as topology becomes sparser.

In (Zhu et al., 2023), loss landscapes are provided as a qualitative demonstration for generalization by reflecting the relationship between the loss and perturbations around the converged parameters. However, in MARL, it is not feasible to compare loss landscapes directly across algorithms due to the dynamic nature of training data (Appendix A.4 Figure 8a). Therefore, we employ the return landscape as a proxy for analyzing landscape flatness, shown in Figure 5a. We also provide the corresponding return histograms (see Appendix A.4, Figure 9a), offering an alternative perspective on the flatness of the return landscapes.

Visually, the reward landscapes for the PS and PS with entropy regularization baselines show a significant drop along the negative $\beta$ direction, indicating that these policies are highly sensitive to perturbations. In contrast, for DecPG, this drop gradually vanishes as $k$ decreases from 8 to 2, eventually resulting in an almost flat landscape. This observation suggests that DecPG with sparser topologies produces more robust policies, as their performance is less sensitive to parameter perturbations.

To sum up, the train and test performance gaps between DecPG and PS confirm that PS suffers from suboptimal convergence and generalization, just like CSGD. In contrast, DecPG, based on DSGD, exhibits clear improvements in exploration, convergence, and generalization over the PS baselines, proving itself as an effective solution to these problems. The noise inherent in DSGD implicitly encourages exploration, which is more effective than entropy regularization in most scenarios. This noise guides the policy toward flatter and more robust optima (as evidenced by the generalization gaps and return landscapes), leading to better convergence and generalization. Regarding the choice of topology sparsity, although no universal pattern exists, the results suggest that a moderate level is generally more reliable, with a spectral gap between 0.5 and 0.7 serving as an empirical guideline.

## 6 Conclusion

In conclusion, this paper proposed a DSGD-based MARL algorithm, DecPG, which leverages decentralized updates to improve generalization and convergence. We conducted an empirical analysis comparing DecPG with its CSGD counterpart, PS, in the MPE Simple Spread and SMAC environments across homogeneous tasks. Our results show that DecPG outperforms PS in both training and test environments in most scenarios, including PS with entropy regularization. This supports our conclusion that PS inherits convergence and generalization issues from CSGD, and that DecPG (DSGD) offers an effective solution.

To further assess generalization, we presented generalization gaps and return landscapes for DecPG and PS baselines. These results illustrate that the regularization effect of DecPG promotes flatter return landscapes, a qualitative illustration of better generalization.

Future work includes extending this research by incorporating more advanced optimizers, such as momentum-based methods like AdamW, which are critical for training modern neural networks like Transformers. Additionally, this paper simulates decentralized training under a single-worker CTDE framework. Therefore, future studies should explore fully distributed training setups to assess the practical training efficiency advantages of DecPG. Finally, since this paper primarily focused on empirical analysis, theoretical justifications are also desirable to better understand the optimization behavior in MARL.

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

# A  Appendix

## A.1  Hyper-parameters

This section provides the hyperparameter configurations used for training the PS, PS+entropy, and DecPG policies across the MPE Simple Spread and SMAC environments. Table 1 summarizes the hyperparameters that are shared across all tasks and methods. Table 2 details the task-specific and algorithm-specific hyperparameters used in SMAC for each method.

For learning rates, we primarily tune over magnitudes [5e-1, 1e-1, 5e-2, 1e-2, 5e-3, 1e-3]. For the 6h_vs_8z and 27m_vs_30m tasks, we additionally test [4e-1, 3e-1, 2e-1, 4e-2, 3e-2, 2e-2], as the default range either led to underfitting or unstable convergence. We adopt a linear learning rate decay from the original lr to the minimum lr.

Table 1: Shared hyperparameter configurations for MPE Simple Spread and SMAC environments.

| Hyperparameter | MPE Simple Spread | SMAC |
|---|---|---|
| Learning rate (lr) | 1e-2 | — |
| Minimum lr | 1e-3 | — |
| lr decay | Linear | Linear |
| Number of mini-batches | 1 | 1 |
| Batch size | 800 | 3200 |
| PPO epochs | 10 | — |
| Clipping parameter | 0.2 | — |
| Entropy coefficient | 1e-3 | 5e-3 |
| Max gradient norm | 10 | 10 |
| Hidden layer size | 64 | 64 |
| Test Environment $c$ | 0.2 | 0.1 |
| Test Environment $\varepsilon$ | 0.2 | 0.01 |
| No. of Evaluation Trials (Train) | 32 | 32 |
| No. of Evaluation Trials (Test) | 1000 | 1000 |

Table 2: Hyperparameter configurations used for SMAC environments across PS, PS+entropy, and DecPG.

| | 5m_vs_6m | | | 10m_vs_11m | | | 27m_vs_30m | | | 6h_vs_8z | | |
|---|---|---|---|---|---|---|---|---|---|---|---|---|
| | PS | PS+entropy | DecPG | PS | PS+entropy | DecPG | PS | PS+entropy | DecPG | PS | PS+entropy | DecPG |
| lr | 1e-2 | 1e-2 | 1e-1 | 1e-2 | 1e-2 | 1e-1 | 1e-1 | 1e-1 | 3e-1 | 1e-1 | 1e-1 | 3e-1 |
| min lr | 1e-3 | 1e-3 | 1e-2 | 1e-3 | 1e-3 | 1e-2 | 1e-2 | 1e-2 | 3e-2 | 1e-2 | 1e-2 | 3e-2 |
| clip param | | 0.05 | | | 0.2 | | | 0.2 | | | 0.2 | |
| ppo epoch | | 10 | | | 10 | | | 5 | | | 5 | |

## A.2  Spectral Gap

The spectral gap of a doubly stochastic matrix $P$ is defined as the difference between its largest eigenvalue (which is 1) and the second-largest eigenvalue in modulus. The formula is given by

$$\text{Spectral Gap} := 1 - \max_{i \geq 2} |\lambda_i|$$

where $\lambda_i$ are the eigenvalues of $P$. An example of the spectral gap values for kNN topologies with 11 agents is shown in Figure 6

## A.3  Policy Entropy

Figure 7 shows the policy entropy of DecPG with different topologies and the PS baselines during training. In the majority of scenarios, DecPG policies maintain higher entropy levels than PS, indicating greater exploration. In contrast, PS with entropy regularization successfully increases exploration, often reaching entropy levels comparable to or higher than those of DecPG.

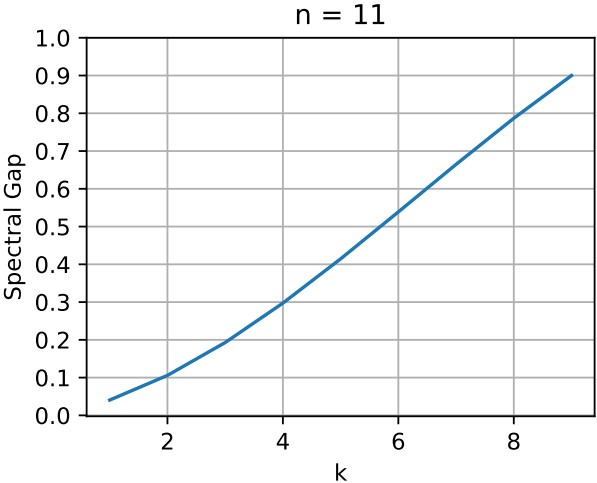

Figure 6: Spectral gaps for $n = 11$ with $k = 1$ to 9

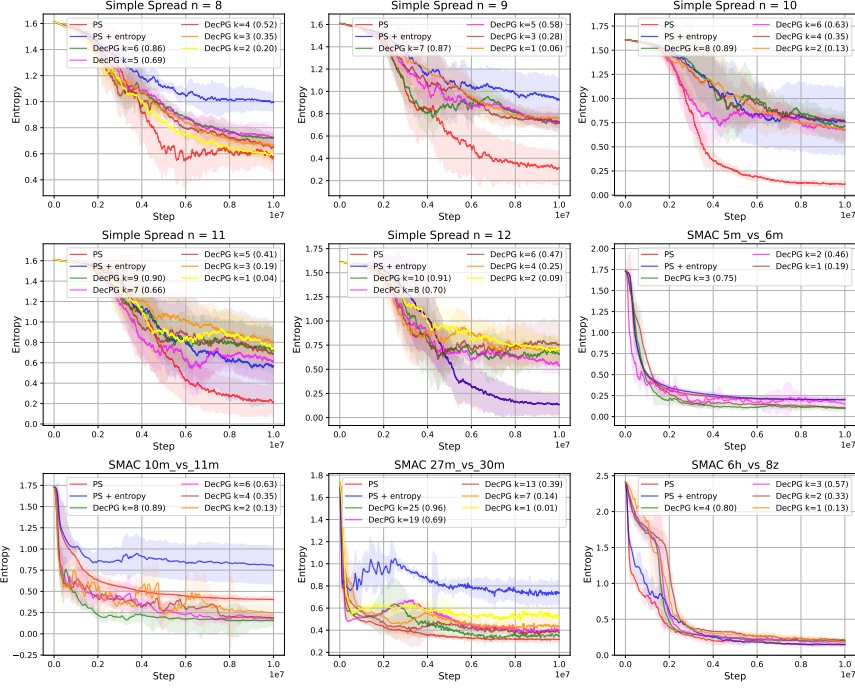

Figure 7: Policy entropy for PS, PS with entropy regularization and DecPG with different topologies.

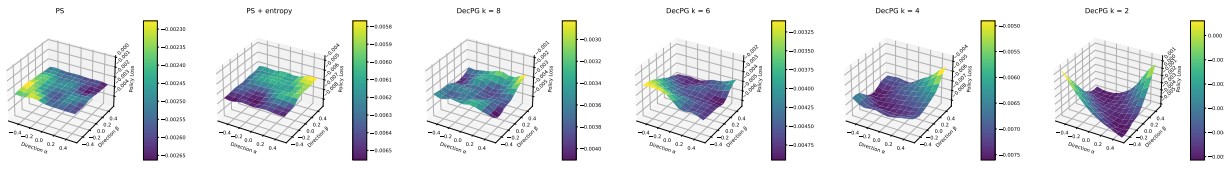

(a) Policy loss landscape surface

Figure 8: Visualization of loss landscapes in MPE Simple Spread $n = 10$, for PS, PS with entropy regularization (PS+entropy), and DecPG with different topologies (DecPG k=x). The figures show 3D surfaces of the loss with respect to perturbations in two orthonormal directions of the parameter space.

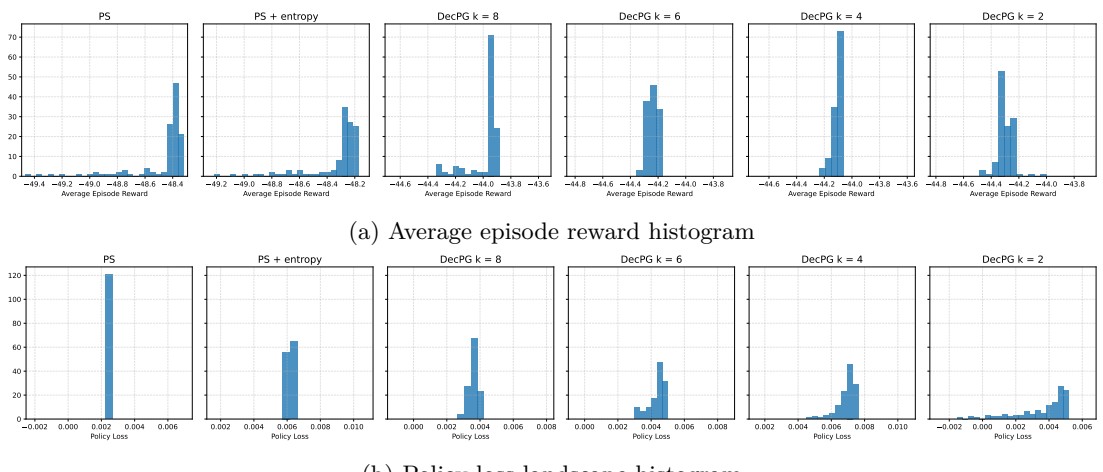

(a) Average episode reward histogram

(b) Policy loss landscape histogram

Figure 9: Visualization of reward and loss landscapes in MPE Simple Spread $n = 10$, for PS, PS with entropy regularization (PS+entropy), and DecPG with different topologies (DecPG k=x). Subfigures **(a)** and **(b)** present the histograms of the same landscape values.

## A.4 Landscapes and Histograms

Figure 8a demonstrates the loss landscapes of PS, PS+entropy, and DecPG with various topologies in Simple Spread n=10. Figure 9 shows the histograms of the reward and loss landscapes corresponding to Figure 8. Longer tails indicate sharper landscapes, while shorter tails suggest flatter ones.

## A.5 Inter-Agent Gradient Disagreement and Update Variance

To verify whether the theoretically predicted decentralization-induced noise actually arises during DecPG training and how it varies across different topology sparsity, we measure the following metrics to quantify and visualize this noise.

We define the inter-agent gradient disagreement $\mathcal{D}$ at a given iteration step $m$ as below:

$$\mathcal{D}_m = \frac{1}{n} \sum_{i=1}^{n} \left\| g_m^i - \bar{g}_m \right\|_2^2, \qquad \text{where } \bar{g}_m = \frac{1}{n} \sum_{i=1}^{n} g_m^i, \tag{1}$$

and $g_m^i = \nabla L_m^i(\theta_m^i)$ denotes the local gradient of agent $i$ on its local loss $L_i$ at iteration $m$.

We define the inter-agent update variance $\mathcal{V}$ at iteration step $m$ as

$$\mathcal{V}_m = \frac{1}{n} \sum_{i=1}^{n} \left\| \Delta\theta_m^i - \overline{\Delta\theta}_m \right\|_2^2, \qquad \text{where } \overline{\Delta\theta}_m = \frac{1}{n} \sum_{i=1}^{n} \Delta\theta_m^i, \tag{2}$$

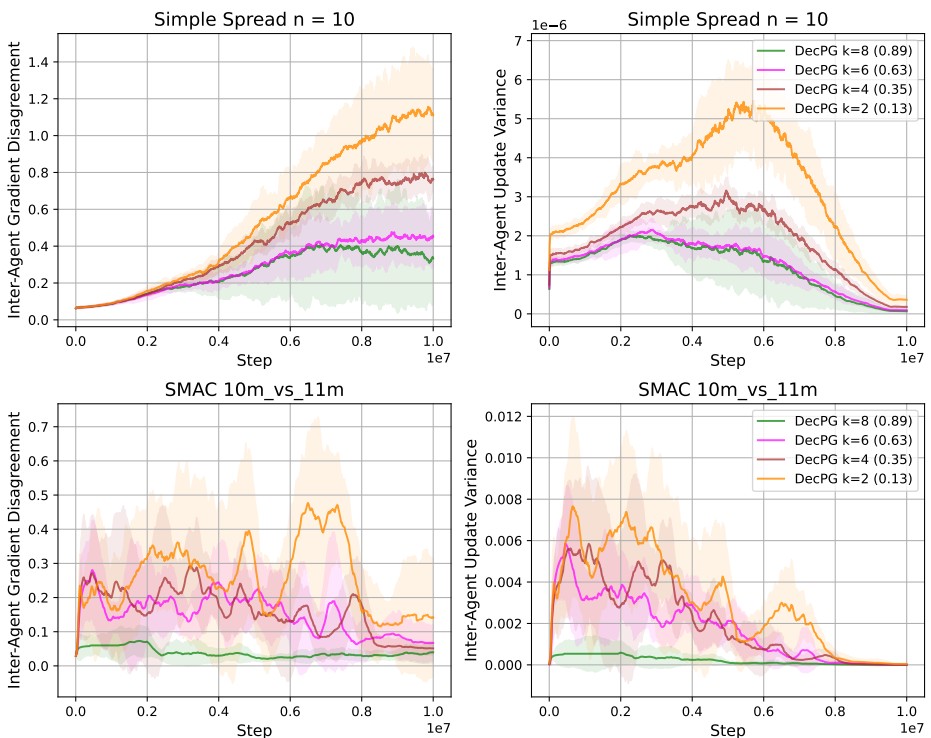

Figure 10: Inter-agent gradient disagreement (left) and inter-agent update variance (right) for DecPG in Simple Spread n=10 (top) and SMAC 10m_vs_11m (bottom).

and $\Delta\theta_m^i = \theta_{m+1}^i - \theta_m^i$ denotes the local parameter update of agent $i$ at iteration $m$.

Figure 10 shows the inter-agent gradient disagreement (left) and update variance (right) of DecPG of different topologies for Simple Spread n = 10 (top) and 10m_vs_11m (bottom) during training. For Simple Spread n = 10, both metrics exhibit clear magnitude differences across topologies. In particular, the topology with $k = 2$ shows the highest levels of inter-agent gradient disagreement and update variance, followed by $k = 4$ and $k = 6$, with $k = 8$ being the smallest. Similarly, in 10m_vs_11m, the $k = 2$ topology yields the largest gradient disagreement and update variance, while $k = 8$ yields the lowest, and $k = 4$ and $k = 6$ remain at comparable intermediate levels. These observations align with theoretical predictions that sparser topologies induce stronger regularization effects (Zhu et al., 2023).

For both tasks, the inter-agent update variance first increases and then decreases toward zero. This trend is partly due to the linear learning rate decay and partly to the gradual diffusion of local parameters across the network, where sparser topologies (smaller $k$) diffuse more slowly, while denser ones (larger $k$) diffuse faster. However, the inter-agent gradient disagreement does not converge to zero by the end of training, particularly in Simple Spread n = 10. This is likely due to the inevitable data heterogeneity across agents. For instance, as training progresses, each agent tends to specialize to some degree, leading to divergent trajectory distributions and consequently greater gradient disagreement than at the start of training.

### A.6 Performance under Different DecPG Evaluation Settings

To understand the characteristics of the final converged DSGD policy, we conduct additional evaluations beyond the standard evaluation using the average policy (denoted as *Average Model (deterministic)* in Table 3), as described in Section 3. First, we evaluate the non-averaged policy, i.e., using each agent's local policy to sample actions for its corresponding agent (denoted as *Local Models* in Table 3). Second, we evaluate each individual agent's policy independently, i.e., using one local policy for the decision-making of all agents. We perform this for every agent and report the mean and standard deviation of performance (denoted as *Indi-*

Table 3: Training and test performance of the four DecPG policy settings for *10m_vs_11m* and *27m_vs_30m*. *Average Model (deterministic)* refers to the standard evaluation setting adopted in this paper. *Average Model (stochastic)* samples actions from the policy distribution. *Local Models* use each agent's local policy to sample actions deterministically. *Individual Local Models* evaluate each local policy deterministically and independently for all agents. For *Individual Local Models*, we report the mean performance and standard deviation across all agents.

| | Average Model (deterministic) | Average Model (stochastic) | Local Models | Individual Local Models |
|---|---|---|---|---|
| 10m_vs_11m k=2 (train) | 12.341 | 11.952 | 12.336 | 12.313(0.073) |
| 10m_vs_11m k=2 (test) | 11.878 | 11.617 | 11.772 | 11.783(0.142) |
| 27m_vs_30m k=1 (train) | 14.309 | 13.032 | 14.241 | 14.234(0.332) |
| 27m_vs_30m k=1 (test) | 14.075 | 12.509 | 14.029 | 13.986(0.317) |

*vidual Local Models* in Table 3). Third, we evaluate the averaged policy stochastically by sampling actions from its distribution (denoted as *Average Model (stochastic)* in Table 3).

We assess both training and test performance for these settings on *10m_vs_11m* (a regular-sized task) and *27m_vs_30m* (a larger-scale task). For both tasks, we use one of the sparsest topologies. Note that it is more difficult for the local models in *27m_vs_30m* to reach consensus than in *10m_vs_11m*, because the sparsest topology in the former is even sparser than in the latter. The results, shown in Table 3, are consistent across all scenarios. The average model performs best, while the local models and individual local models perform slightly lower and at comparable levels. This indicates that the local models remain very close to the averaged model by the end of DecPG training, even when the topology is extremely sparse, suggesting that DecPG behaves similarly to DSGD rather than leading to strong specialization among agents. As expected, the stochastic average model performs worse than its deterministic counterpart in both training and test evaluations.

