# OpenReview forum: "Decentralized Policy Gradients for Optimizing Generalizable Policies in Multi-Agent Reinforcement Learning"
_TMLR — Rejected by TMLR_

### Review · Reviewer_kLoh · 2025-08-29

**Summary Of Contributions:**

This work introduces DecPG, a multi-agent reinforcement algorithm, with parameter-sharing between agents. Based on an analogy between parameter-sharing MARL algorithms and centralized stochastic gradient descent methods (CSGD), the authors apply a successful modification of CSGD and apply it to PS: use a communication topology and partial sharing of gradients.

The authors present results on multiple variants of two families of problems: Simple Spread and the StarCraft Multi-Agent Challenge. The experiments looked at performance against two PS baselines, looking at the effect of communication topology sparseness.

**Additional Comments:**

“All experiments use vanilla SGD as the optimizer with full-batch training.”
How did the authors tune the learning rates? My understanding was that Adam was preferred, unless SGD is carefully tuned to each specific problem.

“All experiments were run for 3 seeds”
 Was each seed only used for 1 run?  That is, are the 95% CI and standard deviation being computed on 3 values?

**Audience:**

Yes

**Audience Explanation:**

People working on multi-agent RL would be interested: having a fairly uncomplicated solution method which generates better and possibly more robust policies, as multi-agent problems do often suffer from low-quality policies, or can be quite poor when other agents in the environment do not act exactly as we expect.

**Claims And Evidence:**

No

**Claims Explanation:**

There are a large number of references to pre-printed work.
Citing pre-printed works seems appropriate to credit existing ideas, or to demonstrate activity within a field. At least to me, it seems inappropriate when presenting a citation as a justification for a statement. For example “CSGD faces several optimization challenges in practice” is followed by 5 references, all of which are pre-prints. There are many other examples: I stopped keeping track by page 3. This leads me to treat large sections of the paper as unsubstantiated claims.

There also seems to be a duplicate reference: Terry 2020a and Terry 2020b are identical, down to the arXiv label.

I’m also not sure that Terry 2020a is an appropriate reference for parameter sharing being possible for heterogeneous agents. For example https://proceedings.neurips.cc/paper/2016/hash/c7635bfd99248a2cdef8249ef7bfbef4-Abstract.html talks about parameter sharing with an agent index on the bottom of page 4, within a general problem setting that is not limited to homogeneous agents. There may certainly be earlier uses, but Terry 2020 is neither the first, not the definitive article on the issue (And following up on the general comment about arxiv references, it is perhaps worth noting that the pre-print seems to have been submitted to ICLR and then withdrawn with a number of negative reviews)

Other related work: decentralized agents with communication often have some sort of agent communication as well. It’s not really my area, so unfortunately I don’t have good suggested citations that represent the area well.

“Furthermore, Ye et al. (2025) show that heterogeneous data can  slow down the convergence and impair generalization. We thus focus primarily on homogeneous problems,  where agents are identical.”
Do the authors intend to limit their claims to homogeneous environments? If so they should make this clear in the abstract and introduction.
If not, this seems to be picking the easiest domains to demonstrate the proposed algorithm, and immediately raises the question “how well does DecPG do in heterogeneous environments?”

Section 4 “ Since the difference in MARL environments across seeds is negligible for evaluating generalization, we adopt a modified version of the training environment for testing, where additional stochasticity is introduced.”
Trying to combine performance and robustness into a single set of results obscures them both : the results with the random actions don’t give the performance on the actual environment, and without results for the base environment without random actions as a comparison, it’s hard to use the random-action results to say something about robustness. So, concretely, I would suggest having results for both with and without the random actions in the test environment.

“These modifications ensure that the test environment does not follow the exact same distribution as the training environment, while still preserving the core task, enabling evaluation of the policy’s ability to generalize without changing the underlying problem.”
Suggest weakening the language here, removing “preserving the core task” and “without changing the underlying problem”
While I agree with the authors that this tests for a kind of robustness, I’m not sure I would agree that the core task remains the same between ‘no error’ and ‘possibility of action error’. There are a number of tasks where I might choose to act very differently if I knew I had a small chance of making entirely random actions.  With multiple mistakes in an episode, I could also end up far enough off the expected distribution of states that it might not be unreasonable for me to be unfamiliar and poorly trained on those states.

Section 5 “In Simple Spread scenarios with n = 9-12, the performance of PS plateaus early during training. In contrast, across all scenarios, DecPG consistently maintains an upward trend and outperforms PS by the end of training.”
While there is certainly a general trend, this seems like a stronger claim than the figure shows, and needs to be reworded more cautiously. For n=9, PS+entropy seems to have a number of the highest values in the zoomed section, and still seems to be roughly on par for n=10. There does seem to be a consistent trend, but for all the simple spread results, both PS+entropy and even PS still look like they have overlapping confidence intervals with the DecPG results.

“Across all scenarios except Simple Spread n = 8, DecPG consistently outperforms the PS baselines in terms of test performance, highlighting the generalization advantage of DSGD-style updates over CSGD”
Another claim that I think needs to be reworded to a more modest or carefully worded claim. Agreeing with the authors statement earlier in section 4, generalization seems to be better described as a function of both the performance in the random-action environment as well as the performance gap from train to test. Just doing better in the test environment doesn’t seem sufficient: an agent might have just done better in the unmodified environment. And there was in fact a general trend of DecPD agents having better performance within the training runs.

“We observe that sparse topologies tend to achieve better test performance”
How do the authors conclude this? Given the error bars and the whole set of 8-12, the results seem consistent with a flat best-fit line and a conclusion that k has no effect. SMAC test performance would seem to generally trend upwards for larger k.
This is maybe repeated again for “observe a decreasing trend in generalization gaps”. Just looking at n=10, this would maybe be more defendable, but with the other choices of n and the variance from Figure 3, this seems like a claim that isn’t obvious from the figures.

**Requested Changes:**

Larger critical changes noted in previous section.

I think there is at least on other field of related work that should be included: decentralized agents with communication often have some sort of agent communication as well. The methods and ideas seem very closely related. It’s not really my area, so unfortunately I don’t have good suggested citations that represent the area well. One early example I do know of is the Foerster et al. 2016 paper I refer to previously, but there are certainly more recent works with decentralized agents making use of limited network topologies.

Small change, Section 4  “We evaluate whether DecPG outperforms entropy regularization, a  commonly used technique in MARL.”
-> “We evaluate whether DecPG outperforms these baseline methods.”   (or something like that: the comparison is against both PS and PS+entropy)

“we assign a small probability ε for an action”
Include the value of epsilon used  (unless I missed it somewhere, I couldn't find it)

---

> ### Author Response · Authors · 2025-11-05
> **Response to Reviewer kLoH Comments (Part 1)**
>
> We sincerely thank the reviewer for their detailed feedback on our paper. We have increased the number of seeds of experiments from 3 to 5 (except for 27m vs 30m, which is still in progress) and revised the paper according to the latest results and the suggestions from the reviews. Below, we address the concerns raised in the review.
>
> # 1. Problems with reference
> We acknowledge that several of our citations refer to preprint works. However, we believe this does not undermine the validity of the points we aim to convey. For example, the references supporting the statement “CSGD faces several optimization challenges in practice” are primarily preprints because these studies focus on empirical investigations of deep neural network training with SGD. Works such as Goyal et al., “Accurate, Large Minibatch SGD: Training ImageNet in 1 Hour”, provide extensive experimental results and valuable insights into how performance is affected by hyperparameter choices, e.g., showing that optimization tends to slow down and even diverge as the batch size increases.
>
> Regarding the citation for PS with ID, we appreciate the reviewer pointing out that citing Terry et al. was inappropriate. We have updated the citation to refer to more representative works in the revised manuscript (Section 2, Page 3).
>
> Goyal et al., “Accurate, Large Minibatch SGD: Training ImageNet in 1 Hour” https://arxiv.org/abs/1706.02677
>
> # 2. Why don't we compare DecPG to the “decentralized agents with communication” algorithm?
> We believe this concern arises from a misunderstanding of the nature of “communication” in DecPG. The "communication" in our work happens when weights are exchanged among neighboring agents to perform a DSGD-style update, which serves purely as an update mechanism (parameter-level “communication”). In contrast, the "communication" in the example paper provided by the reviewer (https://proceedings.neurips.cc/paper/2016/hash/c7635bfd99248a2cdef8249ef7bfbef4-Abstract.html) refers to message passing between agents during the execution phase (policy-level communication), which enables agents to share information for decision-making. Therefore, DecPG is fundamentally different from communication-based methods and is not directly comparable to them.
>
> # 3. Clarification about choosing homogeneous MARL tasks
> In this paper, we intentionally focus on homogeneous MARL tasks, as stated in Sections 1 and 3. This choice is motivated by the fact that the fundamental setup and theoretical guarantees of DSGD are based on homogeneous data across nodes. Nevertheless, homogeneous MARL environments do not necessarily produce homogeneous trajectories across agents. To reduce such discrepancies, we also incorporate CTDE to mitigate non-stationarity and adopt MAPPO to constrain policy updates, making the setup as consistent as possible with the assumptions of DSGD, mentioned in Section 3. We have clarified this motivation more explicitly in the revised paper (Sections 1 and 3).
>
> (to be continued in the next part)

---

> ### Author Response · Authors · 2025-11-05
> **Response to Reviewer kLoH Comments (Part 2)**
>
> # 4. Clarification about the test environment
> In Section 4, we design a dedicated test environment to evaluate the generalizability of algorithms. The conventional evaluation is to evaluate an algorithm on the same environment but with a different random seed from training, which is insufficient for assessing generalization as the difference introduced by changing seeds is often minimal. For example, in Simple Spread, the stochasticity comes from the random initialization of agent and landmark positions, and even with different seeds, those are still sampled from the same underlying distribution. As a result, such evaluations do not meaningfully assess a policy’s robustness. To address this, we design a dedicated test environments that introduce perturbations to observations and actions. It can be easily implemented by modifying the policy's input and output. We treat the evaluation on the same environment with a different seed as the training performance.
>
> Regarding "Trying to combine performance and robustness … hard to use the random-action results to say something about robustness", we believe this stems from a misunderstanding. What the reviewer refers to as “performance” corresponds to what we call training performance, which we report as the performance evaluated on the training environment with different seeds (Figure 2). What R2 refers to as “robustness” aligns with what we call test performance (Figures 3 and 4), i.e., the performance on perturbed environments. In the paper, we analyze training and test performance separately and therefore, they are not conflated.
>
> The reviewer also questions our claim that the test environments “preserve the core task.” Here, we intended to contrast our design with works that alter the environment too drastically, such as changing the number of agents, which effectively turns the evaluation into a few-shot or zero-shot setting, beyond the goal of testing generalization. We agree that the optimal policy in the perturbed test environment may differ from that in the training environment. This is intentional, as it allows us to measure the robustness of learned policies under controlled perturbations, providing a more direct assessment of generalization. We have clarified this motivation in the revised manuscript (section 4).
>
> The perturbations related hyperparameters are stated in Appendix A.1.
>
> # 5. Problems with writing
> We acknowledge that several claims made in Section 5 were insufficiently explained. With the latest results obtained from increasing the number of seeds from 3 to 5, we have systematically rewritten Section 5 to clarify our claims and to focus on explaining the trends in more details. For example, we removed the misleading description that PS/PS+entropy plateau during training and instead focused on a more direct comparison between DecPG and the baselines. In addition, we modified the statement of “sparser topologies lead to better generalization” to a more nuanced analysis of the different trends observed in test performance across tasks. We have also highlighted the theoretical findings from Zhu et al. and point out their connection to the empirical results from our experiments.
>
> # 6. ‘Small change, Section 4 “We evaluate ... DecPG outperforms these baseline methods.”'
> Thank you for pointing this out. We removed this line to improve the overall flow of that paragraph.
>
> (to be continued in the next part)

---

> ### Author Response · Authors · 2025-11-05
> **Response to Reviewer kLoH Comments (Part 3)**
>
> # 7. Clarification about experiment setup (e.g. vanilla SGD, full batch, learning rate)
> In our experiments, we adopt vanilla SGD because the theoretical analyses for advantages of decentralization are derived under this type of optimizer. To ensure consistency with these theoretical findings, we follow the vanilla SGD setting. We acknowledge that exploring more practical optimizers such as AdamW would be valuable for broader applicability. However, enabling DecPG under such optimizers itself is an active research field, which we believe can be separate work.
>
> All algorithms are trained in a full-batch setting. This follows the optimal hyperparameter configuration reported in Yu et al. for MAPPO, where the number of batches is set to 1 for SMAC and Simple Spread.
>
> For learning rates, we primarily tune over magnitudes {5e-1, 1e-1, 5e-2, 1e-2, 5e-3, 1e-3}. For the 6h_vs_8z and 27m_vs_30m tasks, we additionally test {4e-1, 3e-1, 2e-1, 4e-2, 3e-2, 2e-2}, as the default range either led to underfitting or unstable convergence.
>
> We have clarified these details in the revised version of the paper (Sections 3, 4, and Appendix A.1).
>
> Yu, Chao, et al. "The surprising effectiveness of ppo in cooperative multi-agent games." _Advances in neural information processing systems_ 35 (2022): 24611-24624. https://proceedings.neurips.cc/paper_files/paper/2022/hash/9c1535a02f0ce079433344e14d910597-Abstract-Datasets_and_Benchmarks.html
>
> # 8. Problems with seeds
> We have increased the number of random seeds from 3 to 5 for all tasks except 27m_vs_30m, and have updated the corresponding results and analyses in Sections 4 and 5 of the revised paper. The experiments for 27m_vs_30m are still in progress, and we will update the paper with the results once they are completed.
>
> We hope our modifications have addressed most or all of your main concerns. The additional seeds for 27m vs 30m and experiments requested by zxiC are expected to be completed by Friday, November 7, and we will update the manuscript accordingly. We are looking forward to your response.

---

> > ### Comment · Reviewer_kLoh · 2025-11-08
> > **Response**
> >
> > “We acknowledge that several of our citations refer to preprint works. However, we believe this does not undermine the validity of the points we aim to convey.”
> > Something like 13 out of around 34 references seems more like “many” than “several”.
> > There is some difference between citing a pre-print for datasets, credit for ideas or methods, or noting activity in a research area, and citing a pre-print for its conclusions or theoretical arguments. Something like “We compare to … SMAC multiagent challenge” is fine. A statement that XYZ et al. show that ABC is true is something else – directly using a conclusion like that does undermine confidence in any points that rest on it.
> >
> > “We acknowledge that ChatGPT was used to assist with proofreading. In the revised manuscript, we have rewritten sections that appeared overly polished.”
> > This should be mentionedin the manuscript, as noted in https://jmlr.org/tmlr/faq.html
> > “Q: Can I use ChatGPT or other large language model systems to assist in writing my TMLR submission?
> > A: Yes, but TMLR also considers that it is the authors' responsibility to avoid any unforeseen consequences that might come about from using such tools, such as the later discovery that a significant part of the submission's content has been plagiarized (in which case we'd have to reject the paper). Similarly, our expectation is that the ideas, claims and results found in submissions are human-sourced, not produced by such tools.
> > Additionally, transparency is always good, so we also require that authors mention explicitly in their submission that they have used this tool, as a footnote on the first page.”
> >
> > “The reviewer also questions our claim that the test environments “preserve the core task.” Here, we intended to contrast our design with works that alter the environment too drastically, such as changing the number of agents, which effectively turns the evaluation into a few-shot or zero-shot setting, beyond the goal of testing generalization. We agree that the optimal policy in the perturbed test environment may differ from that in the training environment. This is intentional, as it allows us to measure the robustness of learned policies under controlled perturbations, providing a more direct assessment of generalization. We have clarified this motivation in the revised manuscript (section 4).”
> > I understand that the intention was not to drastically alter the task: that’s why my concern was that an epsilon action noise is in fact actually a potentially drastic alteration despite seeming like a small innocuous change. Unless epsilon is small enough to be irrelevant (in which case everything is possibly resting on the observation noise) or has been very carefully tuned.
> > I also don’t see the values of observation noise c and action noise epsilon used in the experiments
> >
> > Resolved —
> > “We believe this concern arises from a misunderstanding of the nature of “communication” in DecPG. The "communication" in our work happens when weights are exchanged among neighboring agents to perform a DSGD-style update, which serves purely as an update mechanism (parameter-level “communication”).”
> > Yes, the communication is different. However, some form of parameter sharing is common, as is a communication topology between agents that can be used in methods which restrict updates to be based on a neighborhood – closer to DecPG than a generic parameter sharing method when the communication actions are removed (but the update topology is preserved).
> > But “I read this at some point” from a reviewer isn’t a concrete reference or an actionable item, so it should not be a requested change, so this issue has been addressed by the author response.
> >
> > “What the reviewer refers to as “performance” corresponds to what we call training performance, which we report as the performance evaluated on the training environment with different seeds (Figure 2). What R2 refers to as “robustness” aligns with what we call test performance (Figures 3 and 4), i.e., the performance on perturbed environments. In the paper, we analyze training and test performance separately and therefore, they are not conflated.”
> > Yes, the added text in section 4 has made the setup and the difference between figures 2-4 more clear.
> >
> >
> > “we have systematically rewritten Section 5 to clarify our claims and to focus on explaining the trends in more details.”
> > Yes, the updates present a better-balanced description of the results.

---

> ### Author Response · Authors · 2025-11-11
> **Follow-up Response to Reviewer kLoh Comments**
>
> We sincerely thank the reviewer for their additional feedback and for acknowledging the revisions made in our previous response. Below, we address the remaining concerns raised in the latest review.
>
> # Clarification about references
> Thank you for pointing out the proportion of preprints in our references. We also found this to be higher than usual and have taken the following actions to reduce it:
> - We noticed that a majority of the preprints we cited were a consequence of downloading citations of conference papers from incorrect BibTeX sources, which caused them to be rendered as preprints. We have now corrected this, and these papers are correctly listed with their official venues.
> - We have replaced some preprints with more representative conference papers for PS.
>
> Through these actions, we have reduced the number of preprints from 13 to 6.
>
> Regarding the remaining preprints, two are from reputable organizations such as Facebook and NVIDIA. Another is a review of CTDE written by Christopher Amato, an associate professor at Northeastern University, which provides a comprehensive overview of the field that we believe is helpful for readers unfamiliar with it. Thus, we consider the sources and reliability of these preprints to be acceptable. Moreover, we do not cite the preprints for unverified methods or theoretical claims, but rather for their empirical observations, such as reported performance issues related to batch sizes. In the case where we previously cited theoretical claims from a preprint, we have revised that to focus solely on the empirical findings (i.e., the Zhang et al. (2021) work in Section 2).
>
> We hope these changes and clarifications address your concerns about retaining the remaining preprints in our references.
>
> # Usage of LLM for proofreading
> Thanks for the instruction. We acknowledge that the ideas, claims and results found in our submission are human-sourced, not produced by AI tools. We have added the clarification in first page footnote.
>
> # Further clarification regarding the test environment
> The concern that the test environment might have had altered the core task drastically is probably a misunderstanding regarding how we randomize the agents’ actions.
>
> When the agents’ actions are randomized at a step, the resulting next states are always produced by legal actions. Specifically, we sample uniformly from each agent’s set of available actions (has now been clarified in the latest revised paper Section 4). The probability of executing the intended action is 1 - epsilon + epsilon / n_a, and the probability of executing any other legal action is epsilon / n_a, where n_a is the number of available actions.
>
> First, this perturbation does not introduce drastic or unrealistic deviations in the trajectories, as the randomized actions are constrained within the legal action sets. Second, the perturbation is independent of both the agents’ chosen actions and the environment state and bounded by the constant epsilon, which means it only adds stochasticity to the transitions, rather than drastically changing the structure of the transition dynamics. The optimal policy therefore remains to select the optimal actions, just as in the original training environment. Third, all algorithms evaluated in our paper are expected to handle such perturbations to varying extents because, at the very least, they were all trained with stochastic policies, meaning that such perturbed trajectories are also possible to occur during training.
>
> Therefore, this perturbation does not change the core task from the training environment, but merely aims to test the policy’s ability to recover from unexpected yet valid state transitions when the system does not perfectly execute the intended action, or equivalently from a different perspective, when the next state distribution is slightly perturbed compared to the training environment.
>
> Regarding the hyperparameters, we used epsilon = 0.2 for Simple Spread and epsilon = 0.01 for SMAC tasks. These values have now been correctly clarified in Appendix A1. The larger epsilon for Simple Spread was necessary because smaller values did not lead to distinguishable generalization gaps due to the simplicity of the tasks compared to SMAC. The reported test returns in Section 5 are averaged over 1000 trials (now clarified in Section 4 and Appendix A1), so the influence of possible extreme outcomes is averaged out.
>
> We hope these clarifications and revisions address your concern. We are looking forward to your response.

---

### Review · Reviewer_zxiC · 2025-08-31

**Summary Of Contributions:**

Summary
This paper replaces parameter-sharing training in multi-agent reinforcement learning with decentralized policy-gradient training (DecPG), uses sparse communication topologies for neighbor averaging/mixing, and examines how topology sparsity and spectral properties relate to training noise and generalization. On MPE and SMAC, it shows that moderate sparsity improves generalization and yields more stable performance than PS/PS+entropy across multiple scenarios.

Strengths

1.Re-examines MARL from a distributed-optimization perspective, clearly linking decentralization to generalization and stating the problem cleanly.

2.Demonstrates gains across multiple MPE/SMAC settings, with generalization and robustness improving in tandem.

3.Implementation is simple, decentralization with neighbor communication swaps into existing PS pipelines with minimal friction.

Weaknesses

1.Optimizer and batching deviate from community practice (vanilla SGD, near full-batch), which may inflate the advantage; add ablations with Adam/momentum and realistic mini-batches.

2.Evaluation relies on a deterministic policy from averaged parameters and is likely optimistic.

3.HAPPO/A2PO and selective sharing (SePS, SNP-PS) are cited, but experiments compare only to PS/PS+entropy.

4.The paper defines the spectral gap but later uses “spectral distance ≤ 0.13.” Unify the term and symbol, and state exactly which graph property the threshold targets.

5.Claims that decentralization-induced noise helps optimization/generalization are not measured; report inter-node gradient disagreement, update variance, and the shares attributable to sampling versus topology.

**Audience:**

Yes

**Audience Explanation:**

1.MARL/CTDE community: Replacing parameter sharing with decentralized policy gradients and systematically examining how topology sparsity relates to generalization offers new insight for this group.

2.Distributed optimization / communication-efficient training: Casting DSGD ideas in policy-gradient form and linking spectral properties to performance can spark discussion on both the algorithmic and systems sides.

3.Graph topology and spectral methods: Explaining training noise and landscape flatness via the spectral gap (or related quantities) is a familiar lens for them but rarely applied in MARL, making it a useful bridge.

4.Robustness and generalization: The systematic sparsity/perturbation sweeps on MPE/SMAC yield practical, reusable empirical takeaways.

**Broader Impact Concerns:**

For details, please see above.

**Claims And Evidence:**

Yes

**Claims Explanation:**

Well supported

1.Across multiple MPE/SMAC scenarios, DecPG shows consistent gains over PS/PS+entropy, and the empirical trend that moderate sparsity improves generalization is backed by continuous experiments (curves over noise levels and topology sparsity, plus visualizations). This class of results is credible.

2.The link between topology sparsity and performance was systematically scanned, not a single-point comparison, with reasonably consistent behavior across tasks.

Evidence is thin

1.Statistical rigor is weak: only three random seeds, no mean±std/CI, no paired tests. Claims of “robust generalization” are under-evidenced.

2.Fairness is questionable: optimizer and batch choices deviate from common practice (vanilla SGD, near full-batch), and DecPG uses much larger learning rates on SMAC. Without equal tuning budgets and sensitivity curves, setup bias cannot be ruled out.

3.Baseline coverage is limited: HAPPO/A2PO and selective sharing (SePS/SNP-PS) are discussed but not included in comparisons; current claims generalize only against PS/PS+entropy.

4.Mechanistic support is missing: the core claim that “decentralization-induced noise aids optimization/generalization” is not quantified—there are no metrics for inter-node gradient disagreement, update variance, or a decomposition into sampling vs. topology effects.

5.Terminology and presentation need consistency (spectral gap vs. spectral distance), and some figures/labels are unclear, which hurts verifiability and repeatability.

**Requested Changes:**

1. In addition to evaluating a deterministic policy from the averaged parameters, please report: (i) per-node policies evaluated independently (mean and distribution), (ii) averaged parameters but stochastic action sampling, and (iii) fully decentralized execution without parameter averaging.
2. Results are based on three random seeds, which is thin for claims about generalization.
3. The manuscript repeatedly claims that decentralization-induced noise aids optimization/generalization but provides no measurement.      Please report inter-node gradient disagreement, update variance, and the fraction attributable to sampling vs. topology.
4. Some passages read like AI-assisted prose.
5. The coordinate axes and scaling are confusing. Replot with consistent axis ranges, labeled units, and a stable legend order across subplots; export vector graphics to avoid artifacts.
6. Unify reference style (journal names, initials, year format), include arXiv version/year, and fix in-text/bibliography ordering mismatches.

---

> ### Author Response · Authors · 2025-11-05
> **Response to Reviewer zxiC Comments (Part 1)**
>
> We sincerely thank the reviewer for their detailed feedback on our paper. We have increased the number of seeds of experiments from 3 to 5 (except for 27m vs 30m, which is still in progress) and revised the paper according to the latest results and the suggestions from the reviews. Below, we address the concerns raised in the review.
>
> # 1. Why choosing SGD with full batch training?
> In our experiments, we adopt vanilla SGD because the theoretical analyses for advantages of decentralization are derived under this optimizer. To ensure consistency with these theoretical findings, we follow the vanilla SGD setting. We acknowledge that exploring more practical optimizers such as AdamW would be valuable for broader applicability. However, enabling DecPG under such optimizers itself is an active research field and requires substantial research work, which we believe goes beyond the scope of our work.
>
> All algorithms are trained in a full-batch setting. This follows the optimal hyperparameter configuration reported in Yu et al. for MAPPO, where the number of batches is set to 1 for SMAC and Simple Spread.
>
> We have clarified these details in the revised version of the paper (Section 4).
>
> Yu, Chao, et al. "The surprising effectiveness of ppo in cooperative multi-agent games." _Advances in neural information processing systems_ 35 (2022): 24611-24624. https://proceedings.neurips.cc/paper_files/paper/2022/hash/9c1535a02f0ce079433344e14d910597-Abstract-Datasets_and_Benchmarks.html
>
>
> # 2. Evaluation relies on a deterministic policy from averaged parameters and is likely optimistic.
> We would like to clarify that the policy is run deterministically only during the evaluation phase, while remains stochastic during training. Training with a stochastic policy—where actions are sampled from the policy distribution—is standard practice and essential for ensuring sufficient exploration. In contrast, evaluation is typically performed deterministically by selecting the action with the highest probability, which is a common procedure in reinforcement learning.
>
> Regarding the use of averaged parameters, we do so for two main reasons. First, this approach aligns with standard practice in DSGD. Second, the theoretical findings from Zhu et al., which we reference when interpreting our results, are derived for the averaged model. Therefore, we intentionally use the averaged model during evaluation to enable a direct application of these theoretical insights.
>
> Zhu, Tongtian, et al. "Decentralized SGD and average-direction SAM are asymptotically equivalent." _International Conference on Machine Learning_. PMLR, 2023. https://proceedings.mlr.press/v202/zhu23e.html
>
> # 3. HAPPO/A2PO and selective sharing (SePS, SNP-PS) are cited, but experiments compare only to PS/PS+entropy.
> In the paper, we do not compare DecPG with parameter-sharing variants such as SePS or non-parameter sharing algorithms like HAPPO, because these methods primarily aim to mitigate inter-agent update conflicts through algorithmic-level designs. In contrast, DecPG introduces a regularization mechanism that operates at the optimization level, focusing on improving generalization rather than coordination. Therefore, DecPG is conceptually orthogonal to those approaches and direct empirical comparison would be less meaningful.
>
> # 4. The paper defines the spectral gap but later uses “spectral distance ≤ 0.13.” Unify the term and symbol, and state exactly which graph property the threshold targets.
> We acknowledge that the mixed use of terminology for describing topologies may have caused confusion. Quantifying a topology solely in terms of k can be difficult, as k depends directly on the number of agents in the task. For example, a spectral gap in the range of 0.5–0.7 may correspond to k = 6 for n = 10, but k = 19 for n = 27. Consequently, in some cases, we refer to topologies using their spectral gap range rather than a specific k value.
>
> In the revised paper, we have addressed this by explaining the connection between spectral gaps and topology sparsity (Section 3) and explicitly providing the intuition of the corresponding k values when referring to spectral gap–based topology sparsity (Section 5.3).
>
> (to be continued in the next part)

---

> ### Author Response · Authors · 2025-11-05
> **Response to Reviewer zxiC Comments (Part 2)**
>
> # 5. Claims that decentralization-induced noise helps optimization/generalization are not measured; report inter-node gradient disagreement, update variance, and the shares attributable to sampling versus topology.
> We agree that this is an essential aspect of our claim. We are conducting additional experiments to measure inter-node gradient disagreement and update variance across different topologies. Since we adopt full-batch training, the sampling variance is zero. We expect to observe that sparser topologies produce larger gradient disagreement and update variance, consistent with the interpretation that decentralization-induced noise acts as a regularizer. Unfortunately, these experiments are still in progress due to limited computational resources, as most have been allocated to increasing the number of random seeds. We will update the paper with the results once they are available.
>
> # 6. In addition to evaluating a deterministic policy from the averaged parameters, please report: (i) per-node policies evaluated independently (mean and distribution), (ii) averaged parameters but stochastic action sampling, and (iii) fully decentralized execution without parameter averaging.
> As mentioned in the response to Question 2, we intentionally use the deterministic averaged model for evaluation. However, we agree that the suggested evaluations would provide additional insight and are planning to include them in the appendix once completed. These experiments are currently in progress, but have not yet been finished due to limited computational resources, as most have been allocated to increasing the number of random seeds. We will report the results once they are ready.
>
> Could you confirm if our understanding of the experiments is right?
> (i)Use each local policy to sample the actions deterministically and report the mean performance
> (ii)Use the averaged policy to stochastically sample the actions
> (iii)Use local policies to sample the actions independently and deterministically
>
>
> # 7. Results are based on three random seeds, which is thin for claims about generalization.
> We have increased the number of random seeds from 3 to 5 for all tasks except 27m_vs_30m, and have updated the corresponding results and analyses in Sections 4 and 5 of the revised paper. The experiments for 27m_vs_30m are still in progress, and we will update the paper with the results once they are completed.
>
> # 8. Some passages read like AI-assisted prose.
> We acknowledge that ChatGPT was used to assist with proofreading. In the revised manuscript, we have rewritten sections that appeared overly polished.
>
> # 9. The coordinate axes and scaling are confusing. Replot with consistent axis ranges, labeled units, and a stable legend order across subplots; export vector graphics to avoid artifacts.
> We have added the color bars for Figure 5, which was missing in the previous version. As far as we can see, we did not find any other figures with incorrect coordinates or scaling. Furthermore, all figures are already provided in PDF format. Could you please specify which figures you identified as having these issues? We will correct them if necessary.
>
> # 10. Unify reference style (journal names, initials, year format), include arXiv version/year, and fix in-text/bibliography ordering mismatches.
> We have double-checked the reference style and did not find any missing journal names, author initials, years, or arXiv versions. Could you please provide an example where you noticed an inconsistency? We have corrected all ordering mismatches between in-text citations and the bibliography in the revised manuscript (Sections 1, 2, and 3).
>
> We hope our modifications have addressed most or all of your main concerns. The additional seeds for 27m vs 30m and experiments requested by zxiC are expected to be completed by Friday, November 7, and we will update the manuscript accordingly. We are looking forward to your response.

---

> ### Author Response · Authors · 2025-11-07
> **Completion of Increased Seeds and Reviewer zxiC-Requested Experiments**
>
> We have completed the additional experiments for increasing seeds from 3 to 5 for all tasks, and the experiments requested by zxiC. The latest results have been updated in the revised paper version 2.
>
> For the training performance, test performance, and generalization gaps, the update does not change the evaluation we made in the previous revision.
>
> Regarding the additional experiments requested by zxiC:
> 1. “Claims that decentralization-induced noise helps optimization/generalization are not measured; report inter-node gradient disagreement, update variance, and the shares attributable to sampling versus topology”, and
> 2. “In addition to evaluating a deterministic policy from the averaged parameters, please report: (i) per-node policies evaluated independently (mean and distribution), (ii) averaged parameters but stochastic action sampling, and (iii) fully decentralized execution without parameter averaging.”
>
> The corresponding results and analyses are presented in Appendix A.5 and A.6.
>
> The findings are summarized as follows.
> For Experiment 1, by plotting the inter-agent gradient disagreement and inter-agent update variance throughout training, we observe a clear relationship between the magnitude of these metrics and topology sparsity: the sparser the topology, the higher the magnitude. This supports Zhu et al.’s theoretical claim that sparser topologies induce stronger noise, leading to a stronger regularization effect.
>
> For Experiment 2, comparing the three additional evaluation settings with the standard setting used in our paper, we find that settings (i) and (iii) yield slightly lower but comparable performance to the standard setting. This indicates that the local policies remain close to the averaged policy by the end of training, consistent with the expected behavior of DSGD-style optimization, and that the averaged policy is indeed the optimal choice for evaluation. Furthermore, the stochastic averaged policy (setting ii) performs the worst among all configurations, empirically confirming our response to Question 2 from zxiC.

---

### Review · Reviewer_recH · 2025-10-21

**Summary Of Contributions:**

**Main Contributions**
The paper aims to make the following contributions:
- Introduction of a Decentralized Policy Gradient (DecPG) method for cooperative multi-agent reinforcement learning (MARL), designed as an alternative to the centralized Parameter Sharing (PS) formulation. The proposed approach seeks to address challenges of convergence to suboptimal policy, failure in convergence when the batches are large, and poor generalization already existing in PS, and improving it's performance.
- Empirical evaluation of DecPG across two multi-agent benchmarks, comparing its convergence, generalization to established baselines of PS and PS with entropy.
- Empirical evaluation of communication graph sparsity on convergence and return landscape.

This seems to be an already tackled problem in the realm of supervised learning, and the authors have tried to bring the solution to the MARL problems.
This seems to be a problem that has already been addressed in the context of supervised learning, and the authors attempt to extend those solutions to multi-agent reinforcement learning (MARL) settings.

**Additional Comments:**

**Overall Assessment:**
The paper presents an interesting idea, but it requires significant polishing in writing and more careful experimental and conceptual clarification to convincingly convey its contributions.

**Audience:**

Yes

**Audience Explanation:**

Yes, the authors provide evidence from prior work highlighting the existing challenges of Parameter Sharing (PS) in MARL, and they propose a method to address these challenges. This approach could be of interest to the MARL research community.

**Broader Impact Concerns:**

-

**Claims And Evidence:**

No

**Claims Explanation:**

Some of the figures are **difficult to interpret**, and several display inconsistent results (specific revisions and elaboration are requested in the next section).
In addition, not all of the claims made in the paper are adequately supported by the presented evidence. While the paper includes numerous plots, there is limited explanation of why the implied result is a consequence of the proposed change or how they substantiate the authors’ conclusions (This could be due to lack to clarity of the text).
Because the performance curves of different algorithms overlap in few plots, the reported results **do not demonstrate clear, statistically significant improvements** of the proposed method across all metrics.

**Requested Changes:**

**Writing and clarity:**
The paper requires careful polishing to clearly convey its main ideas and support it's contributions.

**Missing methodological and elaborative details:**
- Several essential details are missing, which limits both reproducibility of the proposed method and understanding the work along with comparison to prior work:
    + The Parameter sharing (PS) baseline is only described at a high level and it is not clear what the authors refer to. The paper should either (i) cite a precise reference, or (ii) dedicate a subsection explaining the baseline in sufficient detail (including its mathematical formulation) so the comparison is unambiguous.
    + The “PS with entropy” strategy is mentioned but never clearly defined. The paper should specify exactly what this variant entails.
- The authors state that CSGD suffers from certain issues that DSGD solves, and that DecPG similarly addresses those issues for PS. However, not all of these claims are supported by experiments. Specifically, they only evaluate convergence and generalization, while the effect of large batch size is not examined.
- There is no clear answer to ``how and why'' decentralization in DecPG mitigates the alleged problems of the centralized formulation of PS. Some steps appear to be missing in the narrative.

**Figures and Experimental Presentation:**
- Figure 2: The lines are overly cluttered, making it difficult to distinguish algorithms. It may not be necessary to include all sparsity levels for all algorithms. Consider (i) fixing a representative value of k and comparing DecPG with the baselines, and (ii) showing a separate plot for varying k if needed. As shown, the results overlap heavily, and there is no visible improvement in convergence.
- The **color scheme** should be consistent across all figures so that the same algorithm always appears in the same color (e.g., this consistency is missing in Fig. 2).
- Figure 7: Showing standard deviation with sharp lines is visually confusing since they resemble additional algorithms. Use smoothed shading in the same color instead.
- Figure 3:
    + The results overlap heavily, thus, doesn't approve the improvement claim.
    + The decrease in average return with an increasing number of agents is counterintuitive, in particular for the setting of cooperative, homogeneous agents; either this is an error or it needs a clear explanation.
    + There is no consistent pattern when varying k for a fixed n and task. This should be analyzed and discussed.
- Section 5.3 (p. 8, last line): The stated conclusion seems counterintuitive given the cooperative setting and homogeneous nature of the tasks. Please elaborate on why this occurs.
- Figure 5:
   + The plot lacks scaling of the color values—please include that in the figure as well.
   + The role of each dimension, the interpretation of parameters $\alpha$ and $\beta$ need to be explicitly clarified.
- Section 5.4: The intended takeaway from Fig. 5 is unclear; why is it helpful to have a ``flatter return landscape''? This needs to be more clearly elaborated.
- Section 6 (Conclusion): The improvement in generalization gap for DecPG (as seen in Fig. 4) is noted, but the link between this improvement and the analogies to CSGD/DSGD is not substantiated. The paper should explicitly explain why decentralization improves generalization in this specific setting.
- The effect of sparsity is included but not properly analyzed. It remains unclear why or when sparsity helps, or whether it consistently does. The current benchmarks and experiments do not convincingly support the claim that sparsity is beneficial.  For instance, one might  hypthesize that depending on the task and form of reward function, sparsity could be helpful or hurting the performance which needs further investigation.
- Are the evaluation metrics novel and if not, please cite the references. For instance, for generating the test task.
- Several claims in the text lack flow of understanding, and the reader cannot follow the reasoning by reading the text. For instance, when pointing to a paper, you still need to bring the main message and an intuitive support of their claim.

---

> ### Author Response · Authors · 2025-11-05
> **Response to Reviewer recH Comments (Part 1)**
>
> We sincerely thank the reviewer for their detailed feedback on our paper. We have increased the number of seeds of experiments from 3 to 5 (except for 27m vs 30m, which is still in progress) and revised the paper according to the latest results and the suggestions from the reviews. Below, we address the concerns raised in the review.
>
> # 1. What is PS and PS+entropy:
> The Parameter Sharing (PS) baseline refers to a setup where a single neural network is used to represent all agents, and this shared model is updated using trajectories collected from all agents collectively. We have clarified this in Section 2 Page 3 and added the loss function for PS in Section 3 Page 5 of the revised paper.
>
> Gupta et al. Cooperative multi-agent control using deep reinforcement learning. https://link.springer.com/chapter/10.1007/978-3-319-71682-4_5
>
> The PS with entropy baseline (PS+entropy) augments the standard PS loss with an additional entropy regularization term, scaled by a coefficient. This is a common strategy in reinforcement learning to encourage exploration. We have included the explanation and cited the implementation we followed in Section 4 Page 6 of the revised paper.
>
> Yu, Chao, et al. "The surprising effectiveness of ppo in cooperative multi-agent games." _Advances in neural information processing systems_ 35 (2022): 24611-24624. https://proceedings.neurips.cc/paper_files/paper/2022/hash/9c1535a02f0ce079433344e14d910597-Abstract-Datasets_and_Benchmarks.html
>
>
> # 2. Clarification about “CSGD suffers from certain issues that DSGD solves”
> In the paper, we examine whether PS exhibits similar issues to those observed in CSGD and whether DecPG addresses them. By "issues", we mainly refer to the suboptimal convergence and generalization. When discussing the suboptimal convergence observed in prior works, we note that it tends to occur under large-batch training. This remark is included solely to provide context to the related literature and is not a focus of our paper. Nevertheless, our experiments are conducted with full-batch training, following the setup recommended in the MAPPO paper. This setting can be viewed as an indirect evaluation under a large-batch regime. We acknowledge that the term “issues” was not clearly specified in the previous version, which may have led to misunderstanding. In the revision, we removed the “large batch training” in the Introduction and explicitly clarified this by referring to the “issues” directly as suboptimal convergence and generalization (see Section 1 Page 2, Section 2 Page 2, and Section 5 Page 10).
>
> # 3. Why decentralization mitigates problems with convergence and generalization?
> In the paper, we mention that DecPG mitigates the convergence and generalization problems of PS through effect of the decentralization noise. The explanation is supported by the findings from Zhu et al. They prove that the inter-node disagreement noise arising in D-SGD imposes a gradient-smoothing effect on the average model. We interpret the observed improvement in training convergence as a consequence of this smoothing. They further show that D-SGD is asymptotically equivalent to the average-direction Sharpness-Aware Minimization (SAM) algorithm, which acts as a loss-sharpness regularizer. This equivalence implies that D-SGD naturally promotes flatter minima and thus better generalization, consistent with our empirical findings. This work is cited in the Related Work section, and we have further clarified this connection more explicitly in the revised paper (Section 2, Page 3).
>
> Zhu, Tongtian, et al. "Decentralized SGD and average-direction SAM are asymptotically equivalent." _International Conference on Machine Learning_. PMLR, 2023. https://proceedings.mlr.press/v202/zhu23e.html
>
> # 4. Problems about Figure 2 and 7 appearence
> Figures 2 and 7 contained excessive fluctuations and visual clutter. We have applied a running average to smooth the curves and better reveal the trends.
>
> Regarding the colour scheme appearing inconsistent in Figure 2, we would like to clarify our design choices:
> - Lines representing PS are always in red.
> - Lines representing PS+entropy are always in blue.
> - Lines for DecPG with decreasing k values are in the order of green, magenta, brown, orange, and yellow.
> For example, the lines for 5m_vs_6m (k = 2) and 10m_vs_11m (k = 6) are both in magenta because they correspond to the second most densely connected topologies within their respective scenarios. We intentionally do not use identical colours for the same k across different tasks, since the same k value can correspond to very different sparsity levels depending on the number of agents. For instance, k = 3 represents a relatively dense topology in 5m_vs_6m (spectral gap ≈ 0.75) but a much sparser one in Simple Spread n = 11 (spectral gap ≈ 0.19).
>
> (to be continued in next part)

---

> ### Author Response · Authors · 2025-11-05
> **Response to Reviewer recH Comments (Part 2)**
>
> # 5. Problems about Figure 3
>
> ## a. The results overlap heavily
> Figure 3 shows the median test performance over 5 seeds. We acknowledge that the figure contains multiple subplots and error bars, which may make it visually dense. However, upon quantitative inspection, we observe that in most scenarios the median performance of at least one DecPG topology exceeds that of PS and PS+entropy by a margin larger than one standard deviation, which we regard as a clear and consistent improvement. The overlap among DecPG topologies reflects variations in the strength of regularization rather than inconsistency in performance.
>
> If this concern arises from a misunderstanding of the results discussion in the previous version, we kindly advise you to refer to Sections 5.2 and 5.3 of the revised manuscript to see whether the issue remains.
>
> ## b. The decrease in average return with an increasing number of agents is counterintuitive
> The decrease in return with an increasing number of agents is due to the environment design. In MPE Simple Spread, the initialization area for agents and landmarks is fixed. As the number of agents increases, the map becomes more crowded, leading to more frequent collisions during the task. Consequently, tasks with more agents are penalized more often, resulting in lower overall returns.
>
> ## c. There is no consistent pattern when varying k for a fixed n and task
> We acknowledge this observation and have revised Section 5.3 accordingly. In the revision, we have explicitly stated that Figure 3 exhibits increasing, decreasing, and leveling trends, indicating that the behavior of different DecPG topologies depends strongly on the specific task. For instance, tasks like n=12 favor sparser topologies, implying that the underlying problem benefits more from stronger regularization on the optimization landscape's sharpness. In contrast, for 27m vs 30m, excessive regularization may hinder optimization, leading to a drop in test performance. Merging the findings from the three trends, a moderate level of topology sparsity---avoiding the two extremes---appears to be the safer and more robust choice in practice (e.g. with spectral gap between 0.5 and 0.7).
>
> # 6. Section 5.3 (p. 8, last line): The stated conclusion seems counterintuitive given the cooperative setting and homogeneous nature of the tasks
> In Figure 4, we report a decreasing trend in generalization gaps as the topology becomes sparser in the Simple Spread tasks. According to Zhu et al., the sharpness regularization effect of DSGD is related to the sparsity of the communication topology, where a sparser topology imposes stronger regularization on the loss landscape, encouraging convergence to flatter and more generalizable minima. We have clarified this explanation in the revised version of the paper (Section 5.3 Page 9).
>
> Zhu, Tongtian, et al. "Decentralized SGD and average-direction SAM are asymptotically equivalent." _International Conference on Machine Learning_. PMLR, 2023. https://proceedings.mlr.press/v202/zhu23e.html
>
> # 7. Problems with Figure 5:
> ## a. The plot lacks scaling of the color values
> We have added color bars in the revised version.
>
> ## b. The role of each dimension, the interpretation of parameters alpha and beta need to be explicitly clarified.
> This has been clarified in Section 4 and the figure caption. Specifically, the alpha and beta are two orthonormal directions in the policy parameter space and perturbing the policy along those directions.
>
> # 8. In Figure 5, why is it helpful to have a ``flatter return landscape''?
> In Figure 5, we present the return landscapes of the converged PS, PS+entropy, and DecPG algorithms with different k values for the Simple Spread task n = 10, where the landscapes become flatter as k decreases. The return landscape illustrates the relationship between policy return and perturbations around the converged parameters. A flatter return landscape corresponds to a flatter maximum, indicating that the policy performance is less sensitive to parameter changes, which implies greater robustness and better generalization. This observation aligns with the trend in Figure 4, where the generalization gap decreases as k decreases for the same task.
>
> We include this visualization because, in Zhu et al., the loss landscape is used to show that DSGD converges to flatter minima compared to CSGD. However, in MARL, it is not feasible to compare loss landscapes directly across algorithms due to the dynamic nature of training data. Therefore, we employ the return landscape as a proxy for analyzing landscape flatness. We have clarified this reasoning in the revised paper (Section 5.4 Page 10).
>
> (to be continued in the next part)

---

> ### Author Response · Authors · 2025-11-05
> **Response to Reviewer recH Comments (Part 3)**
>
> # 9. Section 6 (Conclusion): The improvement in generalization gap for DecPG (as seen in Fig. 4) is noted, but the link between this improvement and the analogies to CSGD/DSGD is not substantiated. The paper should explicitly explain why decentralization improves generalization in this specific setting.
> PS is a setup where a single neural network is used to represent all agents, and this shared model is updated using trajectories collected from all agents collectively. In Centralized SGD (CSGD), there is only one global model too, which is replicated across all nodes. Each node performs a local gradient descent step, and then the locally updated parameters are averaged and synchronized across all nodes. This procedure is mathematically equivalent to PS, where gradients from all agents are aggregated before performing a single update on the shared model. Therefore, PS and CSGD are considered equivalent in the paper. We’ve added the explicit PS loss in Section 3 of the revised pdf.
>
> In addition, we implement DecPG in a manner that is consistent with DSGD. Specifically, we adopt the Centralized Training with Decentralized Execution (CTDE) paradigm to mitigate the non-stationarity inherent in MARL and use MAPPO to constrain policy updates. To further align our experiments with the CSGD/DSGD setup, we focus on homogeneous tasks and use SGD as the optimizer.
>
> With these measures, the results can be interpreted as a direct result of the theoretical benefits established by Zhu et al. We have emphasized this reasoning more clearly in the revised manuscript, particularly in Sections 1, 2, 3, and 5.
>
> # 10. The effect of sparsity is included but not properly analyzed.
> We acknowledge that in previous version this effect was not sufficiently explained. We have fully rewritten the analyses about different topology sparsity in Section 5.3. Details has been introduced in the answer to question 5c.
>
> # 11. Are the evaluation metrics novel
> Yes, the test environment described in the paper is straightforward: adding gaussian noise to the agents’ observations and randomize agents’ actions given a small possibility at every step.
>
> # 12. Several claims in the text lack flow of understanding
> We have systematically improved the logical flow by emphasizing the theoretical foundations from prior work (Section 2), clarifying the connection between the design of DecPG and DSGD (Section 3), and linking our empirical findings to these theoretical insights (Section 5).
>
> We hope our modifications have addressed most or all of your main concerns. The additional seeds for 27m vs 30m and experiments requested by zxiC are expected to be completed by Friday, November 7, and we will update the manuscript accordingly. We are looking forward to your response.

---

### Decision · Action_Editor_WhVk · 2025-12-23

**Recommendation:** Reject

**Additional Comments:**

This paper, as is, does not meet the acceptance criteria for TMLR because the claims are not backed up by sufficient evidence (either empirical or theoretical). Results published in TMLR do not have to be super-novel, but they do have to hold up to scrutiny, which I am not convinced is the case here.

The authors build their work on a surprising premise: it is possible to increase the performance of the algorithm by imposing a communication / information propagation constraint at training time, while staying in the centralised training paradigm. In other words, the communication constraint serves as a regulariser. While this is a surprising claim, it makes sense to me.

The problem is how the authors set out to build on top of this: the current experiments do not fully support the claim and there is no theory.

In this context, I encourage the authors to resubmit a major revision.

It is up you what to include, but I would suggest starting with a toy example of a dec-pomdp that exaggerates the effect of the regularisation effect due to the communication constraint, ideally making the effect arbitrarily large. I would also suggest looking at alternative (possibly simpler) regularisers to see how well they compare.

Optionally, if you want, you may also try to include a convergence proof, at least for a special case. While deriving performance guarantees for a novel algorithm is hard and typically beyond the scope of one paper, it is good (and easier) to at least show that the iterations converge to *some*  fixpoint i.e. we at least know the algorithm converges to something. I know you have experiments that show a degree of convergence, but they are not enough.

**Audience:**

Yes

**Audience Explanation:**

Multi-agent RL is an important topic for the community. Studying regularisation and generalisation in the centralised training paradigm is a promising dierction.

**Claims And Evidence:**

No

**Claims Explanation:**

The paper claims an improvement in generalization and convergence, due to a new training algorithm.

No theoretical results were provided that look at either one of generalization or convergence.

Empirical evidence provided is not convincing enough (see comments from reviewer recH).

**Resubmission Of Major Revision:**

The authors may consider submitting a major revision at a later time.